

# UV completion of extradimensional Yang-Mills theory for Gauge-Higgs unification

Álvaro Pastor-Gutiérrez[1,2⋆] and Masatoshi Yamada[2,3†]

**1** Max-Planck-Institut für Kernphysik P.O. Box 103980, D 69029, Heidelberg, Germany
**2** Institut für Theoretische Physik, Universität Heidelberg,
Philosophenweg 16, 69120 Heidelberg, Germany
**3** Center for Theoretical Physics and College of Physics, Jilin University,
Changchun 130012, China

⋆ pastor@mpi-hd.mpg.de , † yamada@thphys.uni-heidelberg.de

## Abstract

The $SU(N)$ Yang-Mills theory in $\mathbb{R}^4 \times S^1$ spacetime is studied as a simple toy model of Gauge-Higgs unification. The theory is perturbatively nonrenormalizable but could be formulated as an asymptotically safe theory, namely a nonperturbatively renormalizable theory. We study the fixed point structure of the Yang-Mills theory in $\mathbb{R}^4 \times S^1$ by using the functional renormalization group in the background field approximation. We derive the functional flow equations for the gauge coupling and the background gauge-field potential. There exists a nontrivial fixed point for both couplings at finite compactification radii. At the fixed point, gauge coupling and vacuum energy are both relevant. The renormalization group flow of the gauge coupling describes the smooth transition between the ultraviolet asymptotically safe regime and the strong interacting infrared limit.

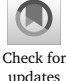

# 1  Introduction

Understanding the dynamics of gauge theories is a challenging problem in theoretical physics. In particular, elucidating their phase structure is a big issue. Gauge theories in four dimensional spacetime ($D = 4$), including quantum chromodynamics (QCD) and similar theories, have been both intensively and extensively studied by various methods, such as Monte-Carlo simulations based on lattice gauge theory and functional methods.

On the other hand, gauge theories in higher dimensional spacetime have not been studied as well as those in $D = 4$. One of typical reasons is that the universe is well-described by four dimensional theories at least up to TeV scales, and there is no experimental evidence for the existence of extra dimensions so far. In addition, gauge theories in $D = 4$ are perturbatively renormalizable and asymptotically free. In terms of the renormalization group, this means that the gauge coupling is relevant at the Gaussian (trivial) fixed point. This fact is crucial for taking a continuum limit and removing lattice artifacts in lattice gauge field theories. In contrast, gauge theories in higher dimensions, e.g. in $D = 5$, are perturbatively nonrenormalizable since the mass dimension of the gauge coupling is negative and the viability of a perturbative expansion over small coupling is lost. Consequently, those are generally regarded as effective field theories with a certain ultraviolet (UV) cutoff and hence predictable only at sufficiently lower energy than the UV cutoff. Nevertheless, gauge theories in higher dimensional spacetime are attractive as both theoretical and phenomenological models in quantum field theory.

From the theoretical side, the notion of renormalizability in quantum field theory can be generalized to nonperturbative theories. In general, most theories in higher dimensional spacetime are *perturbatively nonrenormalizable*, while gauge theories $D > 4$ could be *nonperturbatively renormalizable*. Namely, there could exist a nontrivial fixed point providing a UV-completeness of the theory. This is essential for the asymptotically safe scenario [1,2] (see also Ref. [3] as a review) where a non-trivial fixed point exists in the theory. At such a fixed

point, one can classify couplings by looking at their energy scalings characterized by the critical exponents. A coupling with a positive critical exponent is amplified in the infrared (IR) limit and is called "relevant". An "irrelevant" coupling has a negative critical exponent and converges to the fixed point value in the IR limit. An important fact is that relevant couplings are free parameters, while irrelevant ones are predicted parameters. In general, the number of relevant couplings is finite and thus the theory is predictive for the low energy dynamics. Consequently, the theory becomes *nonperturbatively* renormalizable in the sense that a nontrivial fixed point is generally found by nonperturbative methods.

The existence of a nontrivial UV fixed point in high dimensional gauge theories has been discussed by means of perturbative methods in Refs. [4–8]:[1] The $\epsilon$ expansion was applied for a $SU(N)$ Yang-Mills theory in $4+\epsilon$ dimensional spacetime in order to derive the beta function for the (squared) gauge coupling at the two-loop level as

$$\beta(\tilde{g}) = \epsilon \tilde{g}^2 - \frac{22N}{3}\frac{\tilde{g}^4}{16\pi^2} - \frac{68N^2}{3}\frac{\tilde{g}^6}{(16\pi^2)^2}, \tag{1}$$

where $\tilde{g}^2 = \mu^\epsilon g^2$ is the dimensionless gauge coupling and $\mu$ is the renormalization scale. In addition to the Gaussian (trivial) fixed point $\tilde{g}_*^2 = 0$, characterizing the perturbative theory, the beta function (1) admits a nontrivial UV fixed point,

$$\tilde{g}_*^2 = \frac{16\pi^2}{N}\left(\frac{3}{22}\epsilon - \frac{153}{2662}\epsilon^2 + \cdots\right). \tag{2}$$

The critical exponent at this fixed point is found to be

$$\nu^{-1} = -\frac{d\beta(\tilde{g})}{d\tilde{g}^2}\bigg|_{\tilde{g}^2 = \tilde{g}_*^2} = \epsilon + \frac{51}{121}\epsilon^2 + \cdots. \tag{3}$$

For $\epsilon > 0$, we observe $\nu > 0$, so that the gauge coupling $\tilde{g}$ becomes a relevant coupling. In the limit $\epsilon \to 0$, corresponding to $D = 4$, the nontrivial fixed point (2) merges to the trivial fixed point at which the gauge coupling becomes a marginally relevant coupling to be asymptotically free.

However, the $\epsilon$ expansion is a kind of asymptotic expansion where results for $\epsilon \gtrsim 1$ break down and thus may not be reliable. Hence, in order to establish the asymptotic safety scenario for gauge theories in higher dimensional spacetimes $D > 4$, one requires methods which rely neither on the expansion of spacetimes nor couplings.

On the phenomenological side, higher dimensional gauge theories are the foundation for gauge-Higgs unification models [15–24] which is one of the attractive scenarios for an extension of the Standard Model. The simplest toy model may be a $SU(N)$ Yang-Mills theory in five dimensional spacetime. The extra dimensional component of the gauge field is identified with a scalar field, concretely the Higgs boson. For understanding the dynamics of Gauge-Higgs unification models, the elucidation of the phase structure of gauge theories in higher dimensional spacetime is crucial especially when the extra dimensional direction is compactified to be a circle $S^1$ and an extra gauge field has a nontrivial background configuration. In this case, the gauge symmetry is broken into a subgroup. This noteworthy phenomenon in gauge theories is associated to the Aharonov-Bohm (AB) effect in quantum mechanics or its equivalent, the Hosotani mechanism, in non-Abelian gauge theories [18–20]. Moreover, depending on fermionic degrees of freedom and their representations of the gauge group, the gauge theory tends to have a rich phase structure. See e.g. [20,25,26].

As aforementioned, due to the perturbative nonrenormalizablity of higher dimensional gauge theories, physical quantities in the low energy scale such as the vacuum expectation

---

[1]See also Refs. [9–14] for the renormalization group analysis on higher dimensional gauge theories.

value and the mass of the the Higgs boson are generally UV cutoff dependent. Nevertheless, it is conjectured that the potential determining the vacuum configuration of the extra background gauge field may be free from cutoff dependencies [27, 28]. It is indeed shown in [29–31] that this fact holds at least up to the two-loop level. However, it has been argued recently, in Ref. [32], that the potential is UV sensitive at the four-loop level. See also Ref. [33] for the study on the UV sensitivity in a universal extra dimension model. This means that the vacuum structure is sensitive to higher dimensional operators in perturbative higher loop levels and thus the Gauge-Higgs unification theory fails to predict the low energy physics. Therefore, the UV completion of higher dimensional gauge theories is necessary.

In this paper, we study the $SU(N)$ Yang-Mills theory in $\mathbb{R}^4 \times S^1$ spacetime and its UV completeness in terms of asymptotic safety towards UV complete models of Gauge-Higgs unification. To study the fixed point structure of the $SU(3)$ Yang-Mills theory in $\mathbb{R}^4 \times S^1$ spacetime, we utilize the functional renormalization group [34–37]. See also reviews [38–49]. We derive the (nonperturbative) beta functions for the potential for the background field of an extra gauge field in terms of the AB phases, and for the gauge coupling. It will turn out that the gauge coupling has a nontrivial UV fixed point depending on the compactification radius and then converges to a finite value in the $k \to \infty$ limit, presenting an asymptotically safe theory. Besides, the vacuum energy (the potential independent of AB phases) has a nontrivial UV fixed point as well, so that the potential is free from the UV divergence. These couplings are relevant at the nontrivial UV fixed point, i.e. free parameters in the system.[2] Solving the flow equation for the gauge coupling, it will be clearly seen that the gauge coupling smoothly switches from an asymptotically free behavior to an asymptotically safe one at the energy scale of order of the inverse compactification radius. Then, it converges to a finite value in the UV regime, while in the low energy regime, the gauge coupling becomes a strong coupling which would induce the phase transition to the confinement phase in the $SU(3)$ gauge theory.

However, we cannot address the confinement problem within the current setup. Instead, we demonstrate a mechanism of the confinement by making a simple modification to the beta function of the potential for AB phases. More specifically, we take a gap mass of the gauge fields into account at a certain energy (confinement) scale by hand in order to suppress loop effects of the gauge fields. We will see that the contributions from the ghost fields dominate below the confinement scale and yield nontrivial vacua corresponding to the confinement phase.

This paper is organized as follows: In Section 2 we set up the $SU(N)$ Yang-Mills theory in $\mathbb{R}^4 \times S^1$ spacetime and briefly review the Hosotani mechanism. We introduce the functional renormalization group and explain the idea of asymptotic safety in Section 3. We derive the flow equation for the potential of AB phases in Section 4. The flow equation of the gauge coupling is derived in Section 5. We analyze these flow equations in Section 6. We discuss in Section 7 how a confinement mechanism could be realized by a simple extension. Section 8 is devoted to summarizing this work and to discussing prospects. In Appendix A, the heat kernel method, used to derive the flow equation for the gauge coupling, is explained. Lastly, in Appendix B, the detail of the derivation of the flow equation for the gauge coupling is shown.

## 2 Yang-Mills theory on $\mathbb{R}^4 \times S^1$ spacetime

This section is devoted to a review on the $SU(N)$ Yang-Mills theory in compactified spacetime. We start by defining a Yang-Mills theory in five dimensional spacetime together with a summary of our conventions. Then, we turn to the theory on the compactified spacetime denoted by $\mathbb{R}^4 \times S^1$. For understanding the quantum dynamics of the Yang-Mills theory on $\mathbb{R}^4 \times S^1$, we introduce the Wilson line along with $S^1$ and the Polyakov loop which is defined as the Wilson

---

[2]Strictly speaking, the vacuum energy does not affect the dynamics of the gauge fields.

line traced in color space. The former is a key quantity especially for spontaneous symmetry breaking of gauge symmetry, while the latter is for confinement.

## 2.1 Action

We consider a pure $SU(N)$ Yang-Mills gauge theory in 5 dimensional Euclidean spacetime, where a flat spacetime metric is given by $\eta_{MN} = \text{diag}(1,1,1,1,1)$.

Here and hereafter, we denote 5 dimensional indices by capital letters, e.g. $M, N, \cdots = 0, \ldots, 3, 5$, while greek letters are used to stand for 4 dimensional indices, i.e. $\mu, \nu, \cdots = 0, \ldots, 3$. The Yang-Mills gauge field in 5 dimensional spacetime is then written as $A_M^a = (A_\mu^a, A_5^a)$, where $a, b, \cdots$, denote indices for the adjoint representation of $SU(N)$ whose generators are denoted by $T^a$ and are normalized as $\text{tr}(T^a T^b) = \frac{1}{2}\delta^{ab}$.

The classical action for the Yang-Mills theory in $d = 5$ reads

$$S_{\text{YM}} = \frac{1}{2g^2} \int \text{d}^5 x \, \text{tr}\left[F_{MN} F^{MN}\right]. \tag{4}$$

Here, the field strength of $A_M^a$ and the covariant derivative are given respectively by

$$F_{MN}^a = \partial_M A_N^a - \partial_N A_M^a + f^{abc} A_M^b A_N^c, \tag{5}$$

$$D_M^{ab} = \delta^{ab} \partial_M - f^{abc} A_M^c, \tag{6}$$

with $f^{abc}$ the structure constants associated to $SU(N)$ and $g$ the gauge coupling constant. Note that in 5 dimensional spacetime, the squared gauge coupling $g^2$ has mass dimension $-1$.

## 2.2 Compactification and Kaluza-Klein modes

In five dimensional Euclidean spacetime $\mathbb{R}^5$, spacetime coordinates $x_M$ are noncompact, i.e. $-\infty \le x_M \le \infty$ for $M = 0, \ldots, 3, 5$. The gauge transformation for $A_M = A_M^a T^a$ is given by

$$A_M \to A_M' = U A_M U^{-1} + \frac{i}{g} U \partial_M U^{-1}, \tag{7}$$

where $U = \exp(i\alpha^a(x) T^a)$ are elements of $SU(N)$.

We now define a system in which the $x_5$-direction is compactified so as to be $|x_5| \le R$, where $R$ is a compactification radius. For finite $R$, the system is in $\mathbb{R}^4 \times S^1$ spacetime. Thus, $R$ is a parameter continuously connecting between $\mathbb{R}^5$ ($R \to \infty$) and $\mathbb{R}^4$ ($R \to 0$) spacetimes.

We can generally consider the following boundary condition for the gauge field on $\mathbb{R}^4 \times S^1$:

$$A_M(x, x_5 + 2\pi R) = V A_M(x, x_5) V^{-1}, \tag{8}$$

where $V \in SU(N)$. This boundary condition guarantees the Lagrangian on $S^1$ to be single-valued. The boundary condition for the gauge field transformed by Eq. (7) reads

$$A_M'(x, x_5 + 2\pi R) = V' A_M'(x, x_5) V'^{-1} + iV' \partial_M V'^{-1}, \tag{9}$$

with

$$V' = U(x, x_5 + 2\pi R) V U(x, x_5)^{-1}. \tag{10}$$

In particular, for the gauge transformation by gauge group elements $U_{\text{res}}(x, x_5) \in U(x, x_5)$ satisfying $V' = V$ in Eq. (10), namely

$$V = U_{\text{res}}(x, x_5 + 2\pi R) V U_{\text{res}}(x, x_5)^{-1}, \tag{11}$$

the system on $\mathbb{R}^4 \times S^1$ becomes gauge invariant. In this sense, such a gauge transformation given by $U_{\text{res}}(x, x_5)$ characterizes the "residual" gauge invariance of the system. The choice $V = 1$ does not spoil physical consequences at the quantum level [20], so that hereafter we employ this choice. Then, the gauge field satisfies the periodic condition as can be seen from Eq. (8).

From Eq. (11) with $V = 1$, the residual gauge transformation satisfies

$$U_{\text{res}}(x, x_5 + 2\pi R) = U_{\text{res}}(x, x_5).\tag{12}$$

This concludes that the residual gauge transformation is periodic. Note that one can generalize the transformation (11) such that $V' = ZV$ with $Z$ an element of the center of $SU(N)$, i.e. $Z = \exp(i2\pi k/N)$ with $k = 1,\ldots,N-1$. For the choice $V = 1$, Eq. (10) gives non-periodic transformations

$$U_{\text{res}}(x, x_5 + 2\pi R) = Z U_{\text{res}}(x, x_5).\tag{13}$$

Namely, Eq. (12) is a special case of Eq. (13) with $Z = 1$.

For the gauge field $A_M(x, x_5)$ on $\mathbb{R}^4 \times S^1$ spacetime, one expands the compactified direction as

$$A_M(x, x_5) = \frac{1}{\sqrt{2\pi R}} \sum_{n=-\infty}^{\infty} A_\mu^{(n)}(x) e^{i\Omega_n x_5},\tag{14}$$

where the direction of the extra dimension in momentum space, $p_5$, is given as discretized modes $\Omega_n = n/R$. The summation in Eq. (14) corresponds to the Kaluza-Klein (KK) expansion, representing $n$ the different KK modes. Further, performing the Fourier transformation for $A_M^{(n)}(x)$ in order to move into the 4 dimensional momentum space $p_\mu$, namely

$$A_M^{(n)}(x) = \int \frac{\mathrm{d}^4 p}{(2\pi)^4} A_M^{(n)}(p) e^{ip \cdot x},\tag{15}$$

one has

$$A_M(x, x_5) = \frac{1}{\sqrt{2\pi R}} \sum_{n=-\infty}^{\infty} \int \frac{\mathrm{d}^4 p}{(2\pi)^4} A_M^{(n)}(p) e^{ip \cdot x + i\Omega_n x_5}.\tag{16}$$

For $R \to 0$, the KK frequencies $\Omega_n$ except for $n = 0$ become infinite, so that only lowest mode $n = 0$ contributes to the dynamics. Hence, only $A_M^{(0)}$ are effective degrees of freedom at $R = 0$. On the other hand, the limit $R \to \infty$ yields $\Omega_n \to 0$ for all (finite) KK modes and recovers the continuous momentum $p_5$. Only by performing the full summation all KK modes $A_M^{(n)}$ are taken into account. This fact will be developed in Section 6.

## 2.3 Wilson line and Aharonov-Bohm phase

We introduce the Wilson line which is a key quantity for understanding the gauge dynamics in Yang-Mills theory on $\mathbb{R}^4 \times S^1$. The Wilson line along $S^1$ is defined as

$$W(x) = \mathcal{P} \exp\left\{-i \int_0^{2\pi R} \mathrm{d}x_5 A_5(x, x_5)\right\},\tag{17}$$

where $\mathcal{P}$ is the path ordering and $A_5 = A_5^a T^a$. The residual gauge transformation (12) gives

$$\begin{aligned}W(x) &\to U_{\text{res}}(x, 0) W(x) U_{\text{res}}(x, 2\pi R)^{-1} \\ &= U_{\text{res}}(x, 0) W(x) U_{\text{res}}(x, 0)^{-1}.\end{aligned}\tag{18}$$

Hence, the eigenvalues of $W$ are gauge invariant quantities. We here denote a set of their eigenvalues by $\theta_H \equiv \{\theta_i\}$ $(i = 1, \ldots, N)$. From the fact that the determinant of gauge group elements is unity, the phases $\theta_i$ have to satisfy $\sum_{i=1}^{N} \theta_i = 0 \pmod{2\pi}$. These phases correspond to AB phases in $SU(N)$ gauge theory on $\mathbb{R}^4 \times S^1$. In the next subsection, we see that the AB phases play an important role for the breaking of gauge symmetry.

Another important quantity is the Polyakov loop (in the fundamental representation) which is defined by

$$P = \frac{1}{N} \mathrm{tr}\, W\,, \tag{19}$$

where the trace acts on the fundamental representation space of $SU(N)$. This is transformed under the residual gauge transformation (13) with $Z \neq 1$ as $P \to P' = ZP$. The Polyakov loop is a gauge invariant object, so that it plays a role of an exact order parameter for spontaneous symmetry breaking of the $Z_N$ center symmetry in pure Yang-Mills theories. Moreover, the expectation value of $P$ is written as $\langle P \rangle = \exp(-RF_q)$ where $F_q$ is the free energy of a static massive quark (spectator) at a spacial position. In the confinement phase, $F_q$ has to be infinite and thus we observe $\langle P \rangle = 0$. On the other hand, a finite value of $F_q$ entails $\langle P \rangle \neq 0$ in the deconfinement phase. Hence, the spontaneous symmetry breaking of the $Z_N$ center symmetry corresponds to deconfinement transition.

## 2.4 Hosotani mechanism

Here, we discuss a consequence of a certain nontrivial configuration of $\theta_i$. In this subsection, we see that the phase configurations being $\theta_i \neq \theta_j$ could break the $SU(N)$ gauge symmetry. This mechanism is known as the Hosotani mechanism [18–20], see also Refs. [21, 22]. As a similar system, $D = 4$ YM theories at finite temperature (corresponding to $\mathbb{R}^3 \times S^1$) have been discussed in Refs. [50–60].

We start by assuming that a vacuum state of the system takes $\langle A_\mu \rangle = 0$, $\langle A_5 \rangle \neq 0$. To evaluate quantum fluctuations around such a vacuum, it is convenient to expand the gauge field into the background $\bar{A}_M$ and the fluctuation field $a_M$, i.e.

$$A_M = A_M^a T^a = \bar{A}_M + a_M\,. \tag{20}$$

We assume that the background field takes the following form:

$$\bar{A}_\mu = 0 \qquad (M = \mu = 0, \ldots, 3),$$

$$\bar{A}_5 = \frac{1}{2\pi R} \begin{pmatrix} \theta_1 & & \\ & \ddots & \\ & & \theta_N \end{pmatrix}, \quad \sum_{i=1}^{N} \theta_i = 0 \quad (M = 5)\,. \tag{21}$$

For constant $\theta_i$, the field strength of $\bar{A}_M$ vanishes, i.e. $\bar{F}_{MN} = 0$ and then one has

$$\frac{1}{2} \mathrm{tr}\, \bar{F}_{MN} \bar{F}^{MN} = 0\,. \tag{22}$$

Consider here the Wilson line for the background (21),

$$\bar{W} = \mathcal{P} \exp \left\{ -i \int_0^{2\pi R} \mathrm{d}x_5\, \bar{A}_5(x, x_5) \right\}\,. \tag{23}$$

For a certain configuration of the AB phases, we observe $\bar{W} \neq \mathbb{1}_3$ or correspondingly nontrivial configurations of AB phases. Hence, the $SU(N)$ symmetry is broken into a subgroup $\mathcal{H}_{\mathrm{sym}}$ with

generators satisfying $[T^a, \bar{W}] = 0$, while some generators satisfying $[T^a, \bar{W}] \neq 0$ represent the broken symmetry.

As an example, let us consider a $SU(3)$ gauge theory. There are three constant angles $[\theta_1, \theta_2, \theta_3$ (with $\sum_{i=1}^{3} \theta_i = 0 \mod 2\pi)]$ and then the Wilson line and the Polyakov loop read

$$\bar{W} = \text{diag}\left(e^{i\theta_1}, e^{i\theta_2}, e^{i\theta_3}\right), \tag{24}$$

$$P = \frac{1}{3}\left(e^{i\theta_1} + e^{i\theta_2} + e^{i\theta_3}\right). \tag{25}$$

One can in general write down three different configurations of AB phases: (i) $\theta_1 = \theta_2 = \theta_3 = 0, \pm\frac{2}{3}\pi$, for such configurations, $W$ is proportional to $I$, so that for all Gell-Mann matrices $T^a$, one has $[T^a, \bar{W}] = 0$, i.e. $SU(3)$ symmetry is not broken. (ii) $\theta_1 = \theta_2 = \alpha$ and $\theta_3 = -2\alpha$ ($\alpha \neq 0$). In this case, we observe that $[T^a, \bar{W}] = 0$ only for $a = 1, 2, 3, 8$ which indicates $SU(3) \to SU(2) \times U(1)$. (iii) $\theta_1 = \beta_1$, $\theta_2 = \beta_2$ and $\theta_3 = -\beta_1 - \beta_2$ ($\beta_1 \neq \beta_2$). This configuration shows that only $T^3$ and $T^8$ commute with $\bar{W}$, namely $SU(3) \to U(1) \times U(1)$. These possible symmetry breaking patterns are summarized in Table 1.[3]

Studies of the Hosotani mechanism have been done by means of lattice Monte-Carlo simulations of gauge theories in $\mathbb{R}^3 \times S^1$ in which the continuum limit is manifest [26]. The terminologies of the phases and the configurations were given in Ref. [62]. In particular, AB phases in the confined phase called by $X$ obey the Haar measure and give $P = 0$. Their distribution is similar to the reconfined phase denoted by $C$ in Table 1. AB phases in $X$ would be uniformly and randomly distributed when normalized by the distribution of $\theta_i$ generated by the Haar measure.[4]

As mentioned in subsection 2.3, the confinement-deconfiment phase can be judged by the Polyakov loop. By setting $\alpha = \pi/3$, $\beta_1 = 0$ and $\beta_2 = 2\pi/3$, we plot the distribution of the Polyakov loop (25) in Fig. 1 in accordance with the configurations listed in Table 1. The orange lines are obtained by setting $\theta_1 = \theta_2 = \theta_3 = \varphi$ and varying $\varphi$ from 0 to $2\pi$.

In pure $SU(N)$ Yang-Mills theories, it has been shown by perturbation theory [20] and lattice simulations in $\mathbb{R}^3 \times S^1$ spacetime [26, 62] that in the small gauge coupling regime, the $SU(N)$ symmetry is not spontaneously broken, i.e. the symmetric vacua $\theta_1 = \cdots = \theta_N = 0$ or $\pm 2\pi/N$ are observed as global minima of the effective potential for AB phases. The present functional renormalization group approach will lead to the same result in the $N = 3$ case as of those earlier works and will be presented in Section 4.4.

## 3 Functional flow for Yang-Mills theory

In this section, we set up the functional renormalization group framework for extradimensional Yang-Mills theories. The main objects to be derived are the full propagators for gauge, scalar and ghost fields with an $R_\xi$-gauge fixing action. We perform our computations with the background field method.

---

[3]In the perturbation theory and the functional methods, the gauge fixing is required in order to define the propagators of the gauge fields. Then, we observe that gauge symmetry breaking takes place as summarized in Table 1. On the other hand, in the lattice formulation without the gauge fixing, gauge symmetries are not spontaneously broken thanks to the Elizur's theorem [61]. In this case, gauge symmetries do not play any roles to characterize the phase structure of the system. Instead, the transformations (13) with $Z \neq 1$ act as physical transformations since the physical observables such as the Polyakov loop change. Thus, the physical symmetry group is the quotient group $G/G_0$ where $G$ is the finite group constituted by any value of $Z$ and $G_0$ by $Z = 1$. This quotient $G/G_0$ is isomorphic to the center of $SU(N)$ group and is probed by the Polyakov loop.

[4]Note that in the sense that $P \neq 0$, the deconfined phase with configurations $A_i$ correspond to the "broken" phase, while configurations $C_i$ yields $P = 0$ corresponding to "symmetric" phase. These terminologies are opposite from the symmetry breaking pattern listed in Table 1.

Table 1: Configurations of AB phases and symmetry breaking patters in $SU(3)$ due to the Hosotani mechanism [26, 62], i.e. $SU(3) \rightarrow \mathcal{H}_{\text{sym}}$, where $\alpha \neq 0$, $\beta_1 \neq \beta_2, -\frac{1}{2}\beta_2, -2\beta_2$. Phases in the confined configuration $X$ obey the Haar-measure distribution which is similar to that in the reconfined phase such that $P = 0$. Hence, if the distribution of AB phases in $X$ is normalized by that of the Haar-measure, AB phases in $X$ would be uniformly and randomly distributed. (See Ref. [26].)

| $\mathcal{H}_{\text{sym}}$/Phase | config. | $\theta_1$ | $\theta_2$ | $\theta_3$ |
|---|---|---|---|---|
| $SU(3)$/confined | $X$ | — | — | — |
| $SU(3)$/deconfined | $A_1$ | 0 | 0 | 0 |
| | $A_2$ | $\frac{2}{3}\pi$ | $\frac{2}{3}\pi$ | $\frac{2}{3}\pi$ |
| | $A_3$ | $-\frac{2}{3}\pi$ | $-\frac{2}{3}\pi$ | $-\frac{2}{3}\pi$ |
| $SU(2) \times U(1)$/split | $B_1$ | $\alpha$ | $\alpha$ | $-2\alpha$ |
| | $B_2$ | $\alpha$ | $-2\alpha$ | $\alpha$ |
| | $B_3$ | $-2\alpha$ | $\alpha$ | $\alpha$ |
| $U(1) \times U(1)$/reconfined | $C_1$ | $\beta_1$ | $\beta_2$ | $-\beta_1-\beta_2$ |
| | $C_2$ | $\beta_1$ | $-\beta_1-\beta_2$ | $\beta_2$ |
| | $C_3$ | $-\beta_1-\beta_2$ | $\beta_1$ | $\beta_2$ |

To close this section, we briefly review the basic idea of asymptotic safety. In particular, we focus on the important notion of "relevant" coupling.

## 3.1 Functional renormalization group

A central object in quantum field theory is the one-particle irreducible effective action $\Gamma$. The functional renormalization group provides a tool to obtain $\Gamma$ as the coarse-graining of quantum fluctuations. This can be realized by integrating out quantum fluctuations with higher momentum modes $|p| > k$ and defines the scale-dependent effective (average) action $\Gamma_k$. Then the coarse-graining procedure, i.e. the scale dependence of $\Gamma_k$ is described by a functional differential equation. One of its forms is the Wetterich equation [34]

$$\partial_t \Gamma_k = \frac{1}{2} \text{Tr} \left[ \left( \Gamma_k^{(2)} + \mathcal{R}_k \right)^{-1} \partial_t \mathcal{R}_k \right]. \tag{26}$$

Here, $\partial_t = k \, \partial_k$ is the dimensionless scale derivative, $\Gamma_k^{(2)}$ is the full two-point correlation function obtained by taking the second order functional derivative for $\Gamma_k$ and $\mathcal{R}_k$ is the regulator function. Tr denotes the functional trace acting on all internal spaces where the fields are defined.

In this work, we employ the functional renormalization group equation (26) to analyze the effective action

$$\Gamma_k = \int \mathrm{d}^5 x \, V_{5D}(\theta_H) + \frac{Z_k}{2g^2} \int \mathrm{d}^5 x \, \text{tr} \, F_{MN} F^{MN} + S_{\text{gf}} + S_{\text{gh}}, \tag{27}$$

where $V(\theta_H)$ is the potential for the background field $\bar{A}_5$, i.e. is a function of AB phases. Note that $Z_k$ is the dimensionless field renormalization factor for $A_M$. Here and hereafter the

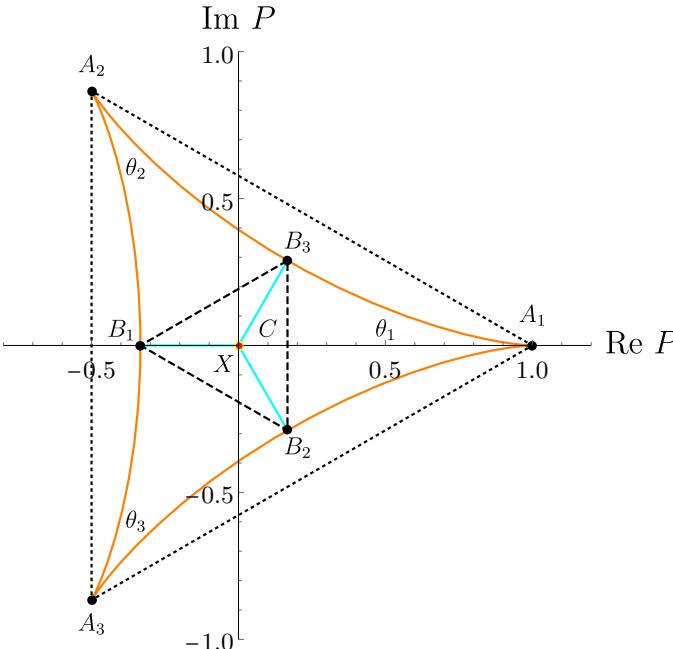

Figure 1: Configurations of the Polyakov loop. We set $\alpha = \pi/3$, $\beta_1 = 0$ and $\beta_2 = 2\pi/3$. The horizontal and vertical axes correspond respectively to the real and imaginary parts of the Polyakov loop. The configurations names are listed in Table 1.

spacetime integral is

$$\int \mathrm{d}^5 x = \int \mathrm{d}^4 x \int_0^{2\pi R} \mathrm{d}x_5, \tag{28}$$

from which if the background field satisfies $\operatorname{tr} F_{MN} F^{MN} = \operatorname{tr} F_{\mu\nu} F^{\mu\nu}$, the potential and the gauge coupling in $D = 4$ are defined as

$$V(\theta_H) = (2\pi R) V_{5D}(\theta_H), \qquad g_{4D}^2 = \frac{g^2}{2\pi R}. \tag{29}$$

We employ the gauge fixing action $S_{\mathrm{gf}}$ and the ghost action $S_{\mathrm{gh}}$, respectively as

$$S_{\mathrm{gf}} = \frac{Z_k}{\xi g^2} \int \mathrm{d}^5 x \operatorname{Tr}[\mathcal{F}(a)]^2, \tag{30}$$

$$S_{\mathrm{gh}} = Z_{\mathrm{gh}} \int \mathrm{d}^5 x \operatorname{Tr}\left[\bar{c} \frac{\delta \mathcal{F}(a^\alpha)}{\delta \alpha} c\right], \tag{31}$$

with $\xi$ the gauge fixing parameter and $Z_{\mathrm{gh}}$ the ghost field renormalization factor. We denote the ghost and anti-ghost fields by $c$ and $\bar{c}$, respectively. For $\mathcal{F}(a)$, we give the $R_\xi$-gauge fixing function

$$\mathcal{F}(a) = \delta^{\mu\nu} \bar{D}_\mu a_\nu + \xi \bar{D}_5 a_5, \tag{32}$$

where $\bar{D}_\mu a_\nu = \partial_\mu a_\nu - i[\bar{A}_\mu, a_\nu]$ is the covariant derivative with the background field. This gauge fixing allows us to cancel the mixing terms between $A_\mu$ and $a_5$. From Eq. (32) and $a_\mu^\alpha = a_\mu + D_\mu \alpha$, one has

$$\frac{\delta \mathcal{F}(a^\alpha)}{\delta \alpha} = \delta^{\mu\nu} \bar{D}_\mu D_\nu + \xi \bar{D}_5 D_5. \tag{33}$$

For the effective action (27), the Wetterich equation is expressed diagrammatically as

$$\partial_t \Gamma_k = \frac{1}{2} \quad + \frac{1}{2} \quad - \quad ,$$

(34)

where curly, dashed and dotted lines represent the 4 dimensional gauge ($M = 0, \ldots, 3$), the extra gauge ($M = 5$) and ghost fields, respectively, and a crossed circle denotes the cutoff insertion, i.e. $\partial_t \mathcal{R}_k$.

Note that in the functional renormalization group, in general, the gauge (or BRST) symmetry is explicitly broken by the introduction of the regulator $R_k$ which requires a modification of the Ward-Takahashi identity. The realization of such a modified identity entails additional contributions like a running gluon mass term to the action at the initial scale in order to cancel gauge symmetry breaking effects due to the regulator. In this paper, we do not take into account such breaking terms. Instead, we will employ the "background field approximation" [36, 63–66] with which the modified Ward-Takahashi identity is satisfied and the background gauge symmetry is realized in the presence of the regulator.

## 3.2 Asymptotic safety

Before we start to calculate the flow equation, we briefly review the basic idea of asymptotic safety. Let us suppose a $d$ dimensional system whose theory space $\Gamma_k$ is spanned by effective operators $\mathcal{O}_i$ with couplings $g_i$:

$$\Gamma_k = \int \mathrm{d}^d x \sum_i g_i \mathcal{O}_i.$$

(35)

Here we denote the mass dimension of $\mathcal{O}_i$ by $d_i$. For the couplings $g_i$, one can compute the renormalization group equations by using Eq. (26) as

$$\partial_t \tilde{g}_i = \beta_i(\{\tilde{g}_j\}),$$

(36)

where $\tilde{g}_i = k^{d-d_i} g_i$ are dimensionless couplings and $\{\tilde{g}_j\}$ denotes a set of dimensionless couplings.

An important notion is the fixed point at which all beta functions (the right-hand side of Eq. (36)) vanish, namely

$$\beta_i(\{\tilde{g}_{j*}\}) = 0, \qquad \text{for all } i.$$

(37)

If such a fixed point $\{\tilde{g}_{i*}\}$ is found, one linearizes Eq. (36) around it as

$$\partial_t \tilde{g}_i \simeq \sum_j \frac{\partial \beta_i}{\partial \tilde{g}_j}\bigg|_{\{\tilde{g}_i\}=\{\tilde{g}_{i*}\}} (\tilde{g}_j - \tilde{g}_{j*}).$$

(38)

By diagonalizing the stability matrix $\frac{\partial \beta_i}{\partial \tilde{g}_j}\big|_{\{\tilde{g}_i\}=\{\tilde{g}_{i*}\}}$ these equations can be solved such that

$$\tilde{g}_i = \tilde{g}_{i*} + \sum_j C_i^j \left(\frac{k}{k_0}\right)^{-\lambda_j},$$

(39)

where $k_0$ is a reference scale and $C_i^j$ is a constant matrix. Here $\lambda_j$ are eigenvalues of the stability matrix $-\frac{\partial \beta_i}{\partial \tilde{g}_j}\big|_{\{\tilde{g}_i\}=\{\tilde{g}_{i*}\}}$ called "critical exponents" and characterize the energy scaling of $\tilde{g}_i$ around the fixed point. Associated couplings with positive $\lambda_i$s are amplified for $k \to 0$

and therefore called "relevant", whereas the "irrelevant" couplings have negative values of $\lambda_i$. The couplings with vanishing critical exponents are "marginal" couplings. In this case, we need to expand the beta functions up to the next order and evaluate whether the next order contribution gives positive contributions to the critical exponents or not. When mixing effects between different operators are negligible, i.e. $C_i^j \approx c_i \delta_i^j$ with constant $c_i$, the critical exponent $\lambda_i$ for a fixed $i$ characterizes the energy scaling for $\tilde{g}_i$. Relevant couplings are free parameters at the fixed point. Then, the finite number of relevant couplings means the theory to be renormalizable. In particular, at the Gaussian fixed point $\tilde{g}_* = 0$, the critical exponents are equivalent to the canonical scaling of the couplings, i.e. $\lambda_i = d - d_i$. The gauge coupling of the Yang-Mills theory in $D = 4$ is a marginally relevant parameter which leads to asymptotic freedom.

In this work, we are especially interested in the renormalization group equation for the dimensionless gauge coupling $\tilde{g}^2$. Its renormalization group equation is schematically given by

$$\partial_t \tilde{g}^2 = \beta_{\tilde{g}^2}(\tilde{g}^2; \bar{R}), \tag{40}$$

where $\bar{R} = kR$ is the dimensionless radius. The explicit definition of $\tilde{g}^2$ and beta function are given in Eq. (74) in Section 5. We look for fixed points by solving $\beta_{\tilde{g}^2}(\tilde{g}^2; \bar{R}) = 0$ as a function of $\bar{R}$. For fixed, finite $\bar{R}$, we will find two fixed points: One is the trivial fixed point $\tilde{g}_*^2 = 0$ for which the canonical scaling term dominates and then the corresponding critical exponent becomes $\lambda_{\tilde{g}^2} = -1$ for $R \to \infty$, while for $R \to 0$ being $\lambda_{\tilde{g}^2} = 0$. The second is the nontrivial fixed point $\tilde{g}_*^2 \neq 0$ at which the critical exponent reads

$$\nu_{\tilde{g}^2}^{-1} = \lambda_{\tilde{g}^2} = -\frac{\partial \beta_{\tilde{g}^2}}{\partial \tilde{g}^2}\bigg|_{\tilde{g}^2 = \tilde{g}_*^2}. \tag{41}$$

We will present the fixed point value and the critical exponent for the gauge coupling as functions of $\bar{R}$ in Section 6. In the asymptotic safety scenario, the vacuum density, i.e. the $\theta_H$-independent part of the potential, should have a finite fixed point too. This is indeed realized in the current study. We will argue this point together with the gauge coupling in Section 6.

## 4 Renormalization group for effective potential

In this section, we derive the flow equation for AB phases $\theta_i$ within the background field approximation [36, 63–66]. See also Refs. [44, 67, 68]. After computing the Hessians for the effective action (27), we derive the flow equation for $V(\theta_H)$. We then show that the vacuum energy $V(0)$ contains a UV divergence, while $V(\theta_H) - V(0)$ becomes finite. We solve the flow equation for $V(\theta_H) - V(0)$ to find its minima at the IR limit ($k \to 0$).

### 4.1 Hessians

To derive the flow equations, we need to compute the Hessian (the two-point function) by performing the second-order functional derivative of $\Gamma_k$, i.e. $\Gamma_k^{(2)} = \frac{\delta^2 \Gamma_k}{\delta \varphi \delta \varphi}$ where $\varphi = (a_\mu, a_5, c)$. The Hessians for the effective action (27) are obtained as

$$\left(\Gamma_k^{(2)}\right)_{a_\mu a_\nu} = \frac{Z_k}{g^2}\left\{\delta^{\mu\nu}\left(-\partial^2 - \bar{D}_5^2\right) + \left(1 - \frac{1}{\xi}\right)\partial^\mu \partial^\nu\right\},$$

$$\left(\Gamma_k^{(2)}\right)_{a_5 a_5} = \frac{Z_k}{g^2}\left(-\partial^2 - \xi \bar{D}_5^2\right),$$

$$\left(\Gamma_k^{(2)}\right)_{\bar{c}c} = Z_{\text{gh}}\left(-\partial^2 - \xi \bar{D}_5^2\right). \tag{42}$$

We note here that $V(\theta_H)$ in Eq. (27) does not contribute to the Hessian since it is a function of the background field. The prefactors $\sim 1/g^2$ come from the overall factor of tr $F_{MN}F^{MN}$ in the effective action (27).

We perform the KK expansion (14) for $\varphi = (a_\mu, a_5, c)$:

$$a_M(x, x_5) = \sum_{n=-\infty}^{\infty} a_M^{(n)}(x) \frac{e^{inx_5/R}}{\sqrt{2\pi R}}, \tag{43}$$

$$c(x, x_5) = \sum_{n=-\infty}^{\infty} c^{(n)}(x) \frac{e^{inx_5/R}}{\sqrt{2\pi R}}, \tag{44}$$

$$\bar{c}(x, x_5) = \sum_{n=-\infty}^{\infty} \bar{c}^{(n)}(x) \frac{e^{inx_5/R}}{\sqrt{2\pi R}}. \tag{45}$$

Inserting these KK expansions, the covariant derivative $\bar{D}_5^2$ acting on $\varphi$ becomes

$$(-\bar{D}_5^2 \varphi)_{ij} = \frac{1}{R^2}\left(n - \frac{\theta_i - \theta_j}{2\pi}\right)^2 \varphi_{ij} \equiv M_{ij,n}^2 \varphi_{ij}, \tag{46}$$

where $i$, $j$ are indices for the fundamental representation of $SU(N)$. Hence, the covariant derivative of the extra dimension acts as a "mass" term for the gauge field. The propagators for $a_\mu^{(n)}(p)$, $a_5^{(n)}(p)$ and the ghost fields in momentum space are given respectively by

$$\Pi_{\mu\nu,ij}^{(n)}(p^2) = \frac{g^2}{Z_k}\left[\frac{\delta_{\mu\nu}}{p^2 + M_{ij,n}^2} + \frac{p_\mu p_\nu}{M_{ij,n}^2}\left(\frac{1}{p^2 + M_{ij,n}^2} - \frac{1}{p^2 + \xi M_{ij,n}^2}\right)\right], \tag{47}$$

$$\Pi_{55,ij}^{(n)}(p^2) = \frac{g^2}{Z_k}\frac{1}{p^2 + \xi M_{ij,n}^2}, \tag{48}$$

$$\Pi_{\text{gh}\,ij}^{(n)}(p^2) = \frac{1}{Z_{\text{gh}}}\frac{1}{p^2 + \xi M_{ij,n}^2}. \tag{49}$$

At this point, we can understand the symmetry breaking patterns from the mass spectra as discussed in Section 2.4. For $\theta_i = 0$ or $\pm\frac{2}{3}\pi$, $M_{ij,n}^2$ has contributions from all KK modes and not from the non-trivial background.

## 4.2 Derivation of flow equation

Let us derive the flow equation of the potential by using the flow equation (26). We first introduce the regulator $\mathcal{R}_k$ such that $-\partial^2$ (or $p^2$) in Eq. (42) is replaced to $P_k(p^2) = p^2 + R_k(p^2)$ by the loop integration, i.e. more specifically, we use the following regulators

$$(\mathcal{R}_k(p^2))^{\mu\nu} = \delta^{\mu\nu}\frac{Z_k}{g^2}R_k(p^2), \quad \text{for } a_\mu^{(n)}, \tag{50}$$

$$\mathcal{R}_k(p^2) = \frac{Z_k}{g^2}R_k(p^2), \quad \text{for } a_5^{(n)}, \tag{51}$$

$$\mathcal{R}_k(p^2) = Z_{\text{gh}}R_k(p^2), \quad \text{for } \bar{c}^{(n)}, c^{(n)}, \tag{52}$$

with a Litim-type cutoff function [69]

$$R_k(p^2) = (k^2 - p^2)\theta(k^2 - p^2). \tag{53}$$

The flow equation (34) in the system reads

$$\partial_t \Gamma_k = \frac{1}{2}\text{Tr}\left[\hat{\Pi}_{\mu\alpha,ij}^{(n)}(P_k)(\partial_t \mathcal{R}_k)^{\alpha\nu}\right] + \frac{1}{2}\text{Tr}\left[\hat{\Pi}_{55,ij}^{(n)}(P_k)\partial_t \mathcal{R}_k\right] - \text{Tr}\left[\hat{\Pi}_{\text{gh}\,ij}^{(n)}(P_k)\partial_t \mathcal{R}_k\right], \tag{54}$$

where $\hat{\Pi}$s denotes the regulated version of the propagators in Eqs. (47) – (49). We define the potential of AB phases as

$$\frac{V(\theta_H)}{2\pi R} = \frac{1}{2\pi R \mathcal{V}_4} \Gamma_k[\bar{A}_5], \tag{55}$$

where $\mathcal{V}_4 = \int \mathrm{d}^4 x$ is the volume in 4 dimensional spacetime and then $2\pi R \mathcal{V}_4 = \int \mathrm{d}^5 x$. The functional trace in Eq. (54) involves summations for gauge group and Lorentz indices and the momenta integration, i.e. for a flow kernel $W(p^2) = W_{ij;n;\mu\nu}(p^2) = \hat{\Pi}^{(n)}_{\mu\alpha,ij}(P_k)(\partial_t \mathcal{R}_k)^{\alpha\nu}$,

$$\mathrm{Tr}[W(p^2)] = \mathcal{V}_4 \sum_{n=-\infty}^{\infty} \int \frac{\mathrm{d}^4 p}{(2\pi)^4} \sum_{i,j=1}^{N} \sum_{\mu=0}^{3} W_{ij;n;\mu}{}^{\mu}(p^2). \tag{56}$$

We show the flow equation in the Feynman gauge $\xi = 1$ by performing the momentum integral and the KK mode summation,

$$\partial_t V(\theta_H) = \frac{5k^4}{2(4\pi)^2}\left(1 - \frac{\eta_g}{6}\right)\mathcal{I}_N(\bar{R}; \theta_H) - \frac{k^4}{(4\pi)^2}\left(1 - \frac{\eta_{\mathrm{gh}}}{6}\right)\mathcal{I}_N(\bar{R}; \theta_H), \tag{57}$$

where $\bar{R} = kR$ is the dimensionless compactification radius, and $\eta_g = -\partial_t \log Z_g$ and $\eta_{\mathrm{gh}} = -\partial_t \log Z_{\mathrm{gh}}$ are the anomalous dimensions of the gauge and ghost fields, respectively.

We have defined the threshold function as

$$\mathcal{I}_N(\bar{R}; \theta_H) = \sum_{i,j=1}^{N} \sum_{n=-\infty}^{\infty} \frac{1}{1 + \bar{M}^2_{ij,n}} = \pi\bar{R} \sum_{i,j=1}^{N} \frac{\sinh(2\pi\bar{R})}{\cosh(2\pi\bar{R}) - \cos(\theta_i - \theta_j)}, \tag{58}$$

where $\bar{M}^2_{ij,n} = M^2_{ij,n}/k^2$. The computation of $\eta_g$ within the background field approximation is given in Section 5.

## 4.3 Independence from UV cutoff

In general, a solution to the differential equation (57) requires a UV cutoff $\Lambda$. However, by subtracting the $\theta_H$-independent part (corresponding to the vacuum energy) from the potential, namely evaluating $V(\theta_H) - V(0)$, the result is free from the UV cutoff. To see this, we consider

$$
\begin{aligned}
&V(\theta_H) - V(0) \\
&= \frac{3}{2(4\pi)^2} \int_0^{\Lambda} \mathrm{d}k\, k^3 (\mathcal{I}_N(\bar{R}; \theta_H) - \mathcal{I}_N(\bar{R}; 0)) \\
&= \frac{-3\pi\bar{R}}{2(4\pi)^2} \sum_{i,j=1}^{N} \int_0^{\Lambda} \mathrm{d}k\, k^4 \frac{\coth(\pi\bar{R}) \sin^2\left(\frac{\theta_i - \theta_j}{2}\right)}{\cosh(2\pi\bar{R}) - \cos(\theta_i - \theta_j)} \\
&= \frac{-3}{2(4\pi)^2} \sum_{i,j=1}^{N} \sin^2\left(\frac{\theta_i - \theta_j}{2}\right) \frac{1}{(\pi R)^4} \int_0^{\pi R\Lambda} \mathrm{d}x \frac{x^4 \coth(x)}{\cosh(2x) - \cos(\theta_i - \theta_j)},
\end{aligned} \tag{59}
$$

where $x = \pi Rk$ and we set $\eta_g = 0$ for simplicity. Hence, the problem is whether the limit $\Lambda \to \infty$ can be taken or not. In other words, we study the convergence of the integral

$$\int_0^{\infty} \mathrm{d}x \frac{x^4 \coth(x)}{\cosh(2x) - a} \qquad (-1 \le a \le 1). \tag{60}$$

Since its integrand increases monotonically for increasing $a$, we see that

$$\int_0^\infty dx \frac{x^4 \coth(x)}{\cosh(2x)+1} = \frac{93}{64}\zeta(5) \leq \int_0^\infty dx \frac{x^4 \coth(x)}{\cosh(2x)-a} \leq \int_0^\infty dx \frac{x^4 \coth(x)}{\cosh(2x)-1} = \frac{3}{2}\zeta(3),$$

(61)

where $\zeta(3) \simeq 1.202$ is the Riemann zeta function of 3. We conclude now that Eq. (59) is finite for $\Lambda \to \infty$. Note that for particular choices of $a$, one has

$$\int_0^\infty dx \frac{x^4 \coth(x)}{\cosh(2x)-a} = \begin{cases} \frac{3}{2}\zeta(3) & (a=1), \\[2mm] \frac{1581}{1024}\zeta(5) & (a=0), \\[2mm] \frac{93}{64}\zeta(5) & (a=-1). \end{cases}$$

(62)

The determination of configurations of AB phases relies on the difference $V(\theta_H) - V(0)$. The vacuum part (the cosmological constant) $V(0)$ is generally divergent. In perturbation theory, the divergence of the $\theta_H$-independent part $V(0)$ in higher dimensional spacetime cannot be subtracted by the standard renormalization, while in the asymptotic safety scenario $V(0)$ could have a nontrivial fixed point, so that it could be an asymptotically safe coupling. Although the vacuum energy does not affect the dynamics of the gauge fields, it is important to see its finiteness as an asymptotically safe feature. We define the dimensionless quantity

$$V(0) = k^4 u(0) = k^4 \mathcal{I}_N(\bar{R};0)\tilde{u}(0).$$

(63)

We reparametrize the potential in such a way that the fifth dimension scales with the dimensionless radius. Taking the $t$-derivative on the both sides of Eq. (63), the flow of $\tilde{u}(0)$ then reads

$$\partial_t \tilde{u}(0) = -d_{\text{eff}}(\bar{R};0)\,\tilde{u}(0) + \frac{1}{k^4 \mathcal{I}_N(\bar{R})}\,\partial_t V(0),$$

(64)

where the last term in the right-hand side is given by Eq. (57) with $\theta_H = 0$. Here we introduce the effective dimension [13] as

$$d_{\text{eff}}(\bar{R};\theta_H) = 4 + \frac{d\log \mathcal{I}_N(\bar{R};\theta_H)}{d\log \bar{R}},$$

(65)

in which we used $\partial_t = \partial/\partial \log k = \partial/\partial \log \bar{R}$. We will investigate the fixed point structure of Eq. (64) in Section 6.

## 4.4  Minimum of the potential

Let us here solve the flow equation for the AB-phase potential in the $N = 3$ case,

$$\partial_t(V(\theta_H) - V(0))$$
$$= \frac{5k^4}{2(4\pi)^2}\left(1 - \frac{\eta_g}{6}\right)(\mathcal{I}_N(\bar{R};\theta_H) - \mathcal{I}_N(\bar{R};0)) - \frac{k^4}{(4\pi)^2}(\mathcal{I}_N(\bar{R};\theta_H) - \mathcal{I}_N(\bar{R};0)),$$

(66)

or equivalently analyze the potential (59). As will be seen in Section 6, the anomalous dimension of the gauge field is bounded $-1 < \eta_g < 0$ and suppressed by a factor $1/6$ and consequently it can be neglected in the flow equation (66).

To solve the differential equation (66), we set $V(\theta_H) - V(0) = 0$ and $\bar{R} = 1$ at $k = \Lambda = 10$ and then evolve the potential $V(\theta_H) - V(0)$ until the low energy scale $k \to 0$. In Fig. 2, the

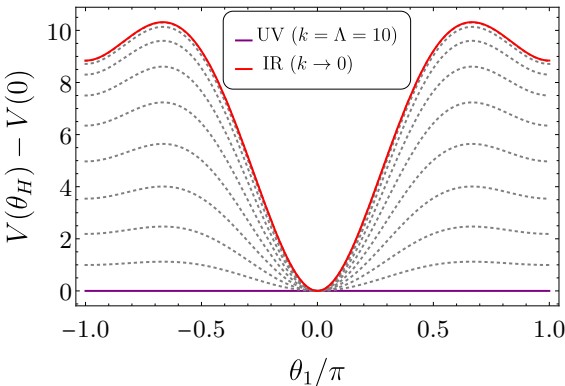

Figure 2: The renormalization group evolution of the potential $V(\theta_H) - V(0)$ at the $\theta_2 = 0$ plane (with $\theta_3 = -\theta_1 - \theta_2$) from the UV to IR scales. The energy units of $k$ are arbitrary. Each dotted line corresponds to the potential at a lowed scale by $\Delta k = 1$.

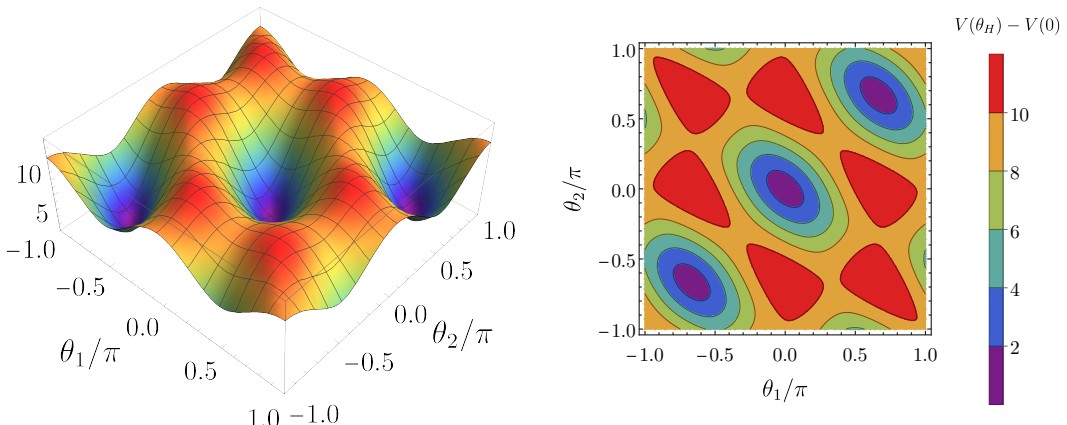

Figure 3: 3 dimensional plot (left) and contour plot (right) of the potential $V(\theta_H) - V(0)$ at the IR scale ($k \to 0$). Note $\theta_3 = -\theta_1 - \theta_2$.

renormalization group evolution of the potential at the $\theta_2 = 0$ plane is presented. Starting from the constant potential at the UV scale, we can see that the potential is deformed and its vacuum structure becomes clearer in lower energy scales. The potential shows a minimum at $\theta_1 = 0$, so that the vacuum is $SU(3)$ symmetric. In order to see this clearly, we plot the potential on the $\theta_1$-$\theta_2$ plane. In Fig. 3, the potential in the IR limit is shown by both 3 dimensional and contour plots. It can be seen that there are vacua at $\theta_1 = \theta_2 = \theta_3 = 0$ and $\theta_1 = \theta_2 = \theta_3 = \pm\frac{2}{3}\pi$. As listed in Table 1, these vacua are $SU(3)$ symmetric and are in a deconfinement phase.

The fact that the $SU(N)$ symmetry is not spontaneously broken in a pure Yang-Mills theory has been actually seen by perturbation theory [20] and lattice simulations of gauge theories on $\mathbb{R}^3 \times S^1$ [26, 62]. We have now confirmed the same conclusion from the functional renormalization group method.

## 5 Flowing gauge coupling

In this section we proceed to derive the flow equation for the gauge coupling. From earlier studies, e.g. [20, 25, 26] and the previous section, it turns out that a pure Yang-Mills theory entails the $SU(N)$ symmetric vacuum $\langle A_5 \rangle = \bar{A}_5 = 0$. Therefore, hereafter we assume a general

constant background $\bar{A}_\mu \neq 0$ and $\bar{A}_5 = 0$ so that

$$\frac{Z_k}{2}\frac{1}{g^2}\int \mathrm{d}^5x\,\mathrm{tr}\big[\bar{F}_{MN}\bar{F}^{MN}\big] = \frac{Z_k}{2}\frac{2\pi R}{g^2}\int \mathrm{d}^4x\,\mathrm{tr}\big[\bar{F}_{\mu\nu}\bar{F}^{\mu\nu}\big].\tag{67}$$

In such a background, the Hessian is obtained by the replacement $\partial_\mu \to \bar{D}_\mu$ and $\bar{D}_5 \to \partial_5$ in Eq. (42) for which we employ the regulator, i.e. $P_k(\bar{\Delta}) = \bar{\Delta} + R_k(\bar{\Delta})$ with $\bar{\Delta} = -\bar{D}^2$. A useful method to obtain the flow equation is the heat kernel technique. Its detailed description is given in Appendix A. For a flow kernel $W(\bar{\Delta}_i)$ with the Laplacian $\bar{\Delta}_i$ acting on spin-$i$ field, the heat kernel expansion results in

$$\mathrm{Tr}_{(i)}[W(\bar{\Delta}_i)] = \frac{\mathcal{V}_4}{(4\pi)^2}\sum_{n=-\infty}^{\infty}\left[b_0^{(i)}Q_2[W] + b_2^{(i)}Q_0[W]\mathrm{tr}\,\bar{F}_{\mu\nu}\bar{F}^{\mu\nu} + \cdots\right].\tag{68}$$

Here $Q_n[W]$ are the threshold functions given by

$$Q_2[W] = \int_0^\infty \mathrm{d}z\,z\,W(z),\tag{69}$$

$$Q_0[W] = W(0),\tag{70}$$

and $b_j^{(i)}$ are the corresponding heat kernel coefficients for the spin representation $(i)$ of the field. The first term on the right-hand side in Eq. (68) corresponds to the beta functions for $V(\theta_H = 0)$. The flow equation for $Z_k$ is thus obtained from

$$\frac{\partial_t Z_k}{2}\frac{2\pi R}{g^2} = \frac{1}{\mathcal{V}_4}\partial_t\Gamma_k\bigg|_{F_{\mu\nu}^2}.\tag{71}$$

The evaluation of the right-hand side is presented in Appendix B. Here we show the result obtained from the flow equation for the gauge coupling

$$\frac{\partial_t Z_k}{2}\frac{2\pi R}{g^2} = \frac{N}{2(4\pi)^2}\left[\frac{20}{3}\left(1-\frac{\eta_g}{2}\right)-\frac{1}{3}\left(1-\frac{\eta_g}{2}\right)+\frac{2}{3}\left(1-\frac{\eta_{\mathrm{gh}}}{2}\right)\right]\frac{1}{N^2}\mathcal{I}_N(\bar{R};\theta_H=0).\tag{72}$$

Defining the dimensionless renormalized gauge coupling

$$\tilde{g}^2 = \frac{Z_k^{-1}g^2}{2\pi R}\frac{\mathcal{I}_N(\bar{R};0)}{N^2} = Z_k^{-1}g_{4\mathrm{D}}^2\frac{\mathcal{I}_N(\bar{R};0)}{N^2},\tag{73}$$

and taking the derivative of both sides in Eq. (73) with respect to $t$ which acts on $\tilde{g}$, $Z_k^{-1}$ and $\bar{R}$, the flow equation for $\tilde{g}$ can be written as

$$\partial_t\tilde{g}^2 = \big(d_{\mathrm{eff}}(\bar{R};0)-4+\eta_g\big)\tilde{g}^2,\tag{74}$$

where $d_{\mathrm{eff}}(\bar{R};\theta_H)$ has been defined in Eq. (65). Here, the anomalous dimension reads

$$\eta_g = -\frac{\partial_t Z_k}{Z_k} = -\frac{N}{(4\pi)^2}\left(7-\frac{19\eta_g}{6}+\frac{\eta_{\mathrm{gh}}}{3}\right)\frac{Z_k^{-1}g^2}{2\pi R}\frac{\mathcal{I}_N(\bar{R};0)}{N^2}.\tag{75}$$

Setting $\eta_g = \eta_{\mathrm{gh}} = 0$ in the right-hand side of Eq. (75) produces the one-loop contribution to the beta function

$$\eta_g = -7N\frac{\tilde{g}^2}{(4\pi)^2}.\tag{76}$$

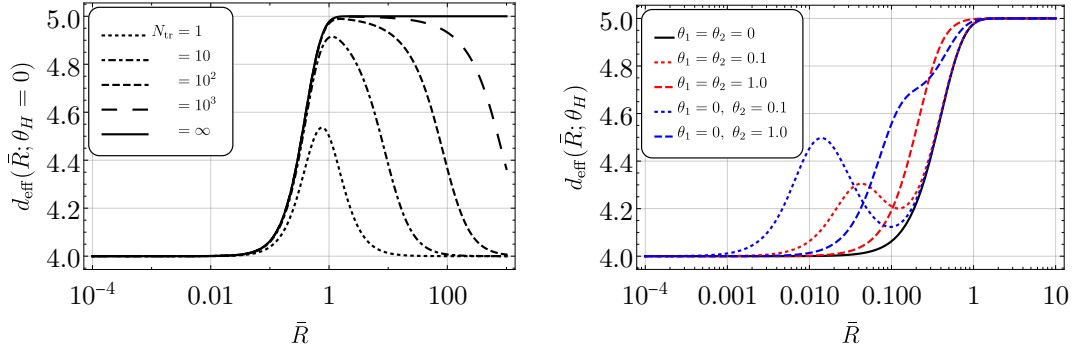

Figure 4: The effective dimension $d_{\text{eff}}$ as a function of the dimensionless compactification radius. Left: Dependence of $d_{\text{eff}}$ on truncation limits on the KK mode summations as $|n| = N_{\text{tr}}$ in the symmetric phase ($\theta_1 = \theta_2 = \theta_3 = 0$). Right: The effective dimension with the full resummation of KK modes for the different AB phases shown in Tabel 1. Note $\theta_3 = -\theta_1 - \theta_2$.

This one-loop result agrees with that of Ref. [33] in which a Yang-Mills theory in universal extra dimensions is studied. Setting only $\eta_{\text{gh}} = 0$ and solving Eq. (75) to $\eta_g$ yield the "resummed" anomalous dimension of the gauge field as

$$\eta_g = -\frac{7N \frac{\tilde{g}^2}{(4\pi)^2}}{1 - \frac{19N}{6} \frac{\tilde{g}^2}{(4\pi)^2}} \,. \tag{77}$$

This corresponds to the loop resummation, i.e. the nonperturbative anomalous dimension. Indeed, expanding it into the polynomial of $\tilde{g}^2$ gives

$$\eta_g = -7N \frac{\tilde{g}^2}{(4\pi)^2} - \frac{133N^2}{6} \frac{\tilde{g}^4}{(4\pi)^4} + \mathcal{O}\left(\tilde{g}^6\right) \,. \tag{78}$$

The first term agrees with the one-loop result given in Eq. (76). We should here note that the one-loop coefficient ($-7$) is different from that ($-22/3$) of Eq. (1) because the present system contains an additional scalar one-loop contribution from $A_5$. Subtracting its contribution from the anomalous dimension by hand actually reproduces the same coefficient as of the one-loop result in Yang-Mills. The two-loop coefficient also differs from Yang-Mills: This is because the background field approximation [36,63–66] misses some contributions from two-loop effects besides the $A_5$ contribution.

# 6 Fixed point structure, critical exponent and renormalization group flow

Before studying the fixed point structure of $\tilde{u}(0)$ and $\tilde{g}^2$, let us see the behavior of the effective dimension $d_{\text{eff}}(\theta_H, \bar{R})$ defined in Eq. (65). The effective dimension is depicted in Fig. 4 for $N = 3$: The left-hand side panel shows $d_{\text{eff}}$ for different truncations for the KK modes, i.e. the KK summation is truncated as $\sum_{n=-N_{\text{tr}}}^{N_{\text{tr}}}$. We see that the effective dimension smoothly interpolates between four and five dimensional spacetimes varying the compactification radius if all KK modes are summed up ($N_{\text{tr}} = \infty$):

$$d_{\text{eff}}(\bar{R}; \theta_H) = \begin{cases} 4 & (\bar{R} \to 0), \\ 5 & (\bar{R} \to \infty). \end{cases} \tag{79}$$

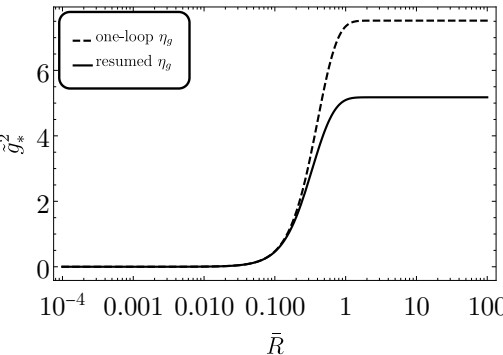 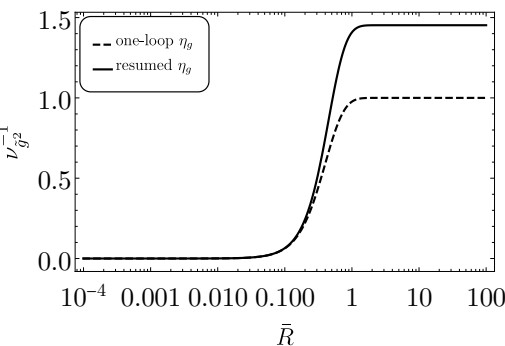

Figure 5: The nontrivial fixed point value $\tilde{g}_*^2$ (left) and the critical exponent $\nu_{\tilde{g}^2}^{-1}$ (right) of the gauge coupling as a function of the dimensionless compactification radius $\bar{R}$.

For finite truncations ($N_{\text{tr}} < \infty$), however, the full dimensional transition is not achieved. This originates from the fact that all KK modes $M_n = n/R$ in $R \to \infty$ become massless and thus contribute to the loop integration. The effective dimension $d_{\text{eff}}(\bar{R} : \theta_H)$ is defined from the threshold function (loop integral) (58), so that it describes the dynamical change of the dimensionality in the system. In the right-hand side panel of Fig. 4, the effective dimension is shown for the different breaking patterns listed in Table 1. We see a nontrivial dependence of the dimensionality on the background field configuration. Such a dependence is observed especially around the compactification radii $0.001 \lesssim \bar{R} \lesssim 1$ for which the transition of dimensionality takes place. Nevertheless, the behavior of Eq. (79) is independent of the vacuum configurations.

We now study the fixed point solutions in the flow equations:

$$\partial_t \tilde{u}(0) = -d_{\text{eff}}(\bar{R};0)\,\tilde{u}(0) + \frac{5}{2(4\pi)^2}\left(1 - \frac{\eta_g}{6}\right) - \frac{1}{(4\pi)^2}\,, \tag{80}$$

$$\partial_t \tilde{g}^2 = \left(d_{\text{eff}}(\bar{R};0) - 4 + \eta_g\right)\tilde{g}^2 \equiv \beta_{\tilde{g}^2}\,, \tag{81}$$

with the effective dimension (65) and the anomalous dimension of the gauge field

$$\eta_g = \begin{cases} -\dfrac{\frac{21}{(4\pi)^2}\tilde{g}^2}{1 - \frac{57}{6(4\pi)^2}\tilde{g}^2} & \text{(resumed)}\,, \\[4mm] -\dfrac{21}{(4\pi)^2}\tilde{g}^2 & \text{(one-loop)}\,, \end{cases} \tag{82}$$

where "one-loop" corresponds to the lowest order of the Taylor series in terms of $\tilde{g}^2$ for the "resumed" anomalous dimension. We set $\eta_{\text{gh}} = 0$ in this analysis. Let us first discuss the nontrivial UV fixed point of the gauge coupling. We find it analytically as

$$\tilde{g}_*^2 = \begin{cases} \dfrac{32\pi^2(\sinh(2\pi\bar{R}) - 2\pi\bar{R})}{61\sinh(2\pi\bar{R}) - 38\pi\bar{R}} & \text{(resumed)}\,, \\[4mm] \dfrac{16\pi^2}{21} - \dfrac{16\pi^2}{21}\dfrac{2\pi\bar{R}}{\sinh(2\pi\bar{R})} & \text{(one-loop)}\,. \end{cases} \tag{83}$$

Fig. 5 exhibits this fixed point value and the critical exponent of the gauge coupling $\tilde{g}^2$ as functions of $\bar{R}$ in the symmetric vacuum $\theta_1 = \theta_2 = \theta_3 = 0$. For $\bar{R} \to \infty$ ($\mathbb{R}^5$ spacetime), the

fixed point value $\tilde{g}_*^2$ is given by

$$\tilde{g}_*^2\big|_{\bar{R}\to\infty} = \begin{cases} \dfrac{32\pi^2}{61} \simeq 5.18 & \text{(resumed)}, \\[4mm] \dfrac{16\pi^2}{21} \simeq 7.52 & \text{(one-loop)}, \end{cases} \tag{84}$$

while for $\bar{R} \to 0$ corresponding to the system in $D = 4$, the fixed point value goes to zero, namely, the gauge coupling converges to the Gaussian fixed point which entails asymptotic freedom of the gauge coupling in $D = 4$. This can be also seen from Fig. 6 in which the behavior of the beta function $\beta_{\tilde{g}^2}$ in Eq. (81) as a function of $\tilde{g}^2$ is depicted. As the compactification radius becomes smaller, the nontrivial fixed point value comes closer to the Gaussian fixed point. For $\bar{R} \to 0$, the nontrivial fixed point merges with the Gaussian fixed point. The critical exponent has been defined in Eq. (41) which reads for Eq. (81) at the nontrivial fixed point as

$$\nu_{\tilde{g}^2}^{-1} = 4 - d_{\text{eff}}(\bar{R}; 0) - \frac{\partial}{\partial \tilde{g}^2}(\eta_g \tilde{g}^2)\Big|_{\tilde{g}^2=\tilde{g}_*^2}. \tag{85}$$

For a finite compactification radius, one has $d_{\text{eff}}(\bar{R}; 0) > 4$, so that the first two terms in the right-hand side of Eq. (85) yield a negative contribution to the critical exponent. Since in the Gaussian fixed point $\tilde{g}_*^2 = 0$, the last term (anomalous dimension) in Eq. (85) vanishes, the gauge coupling becomes irrelevant. This reflects the perturbative nonrenormalizability of higher dimensional Yang-Mills theories. On the other hand, at the nontrivial fixed point (84), a finite positive value of the anomalous dimension is induced and consequently the total value of the critical exponent could be positive. The right-hand side panel of Fig. 5 tells us actually that the critical exponent of the gauge coupling is positive for all compactification radii. Hence, the gauge coupling is a relevant coupling and thus a free parameter. The value of the critical exponent $\nu_{\tilde{g}^2}^{-1}$ especially for $\bar{R} \to \infty$ becomes

$$\nu_{\tilde{g}^2}^{-1} = \begin{cases} \dfrac{61}{42} \simeq 1.45 & \text{(resumed)}, \\[4mm] 1 & \text{(one-loop)}. \end{cases} \tag{86}$$

We next analyze the fixed point and the critical exponent for $\tilde{u}(0)$. To this end, we note that the value of the anomalous dimension $\eta_g$ at the UV fixed point is in $-1 < \eta_g|_{\tilde{g}^2=\tilde{g}_*^2} < 0$ for a finite compactification radius and therefore the effect of $\eta_g$ in the vacuum energy (80) is negligible due to the suppression factor $1/6$. However, if we want to propagate the existence of a non-trivial fixed point in the gauge coupling to the flow of the dimensionless potential, it is necessary to account for its contribution via the anomalous dimension. Also, note that a vanishing anomalous dimension in the computation of the flow of the potential may be confused with the presence of a gaussian fixed point in the $D \to 5$ limit. Consequently, a nontrivial UV fixed point for a certain $\bar{R}$ is obtained

$$u_*^{\text{UV}}(0) = \begin{cases} \dfrac{23}{960\pi^2} + \dfrac{\bar{R}}{240\pi(2\pi\bar{R} - 5\sinh(2\pi\bar{R}))} & \text{(resumed)}, \\[5mm] \dfrac{3}{32\pi^2\left(5 - \frac{2\pi\bar{R}}{\sinh(2\pi\bar{R})}\right)} & (\eta_g = 0), \end{cases} \tag{87}$$

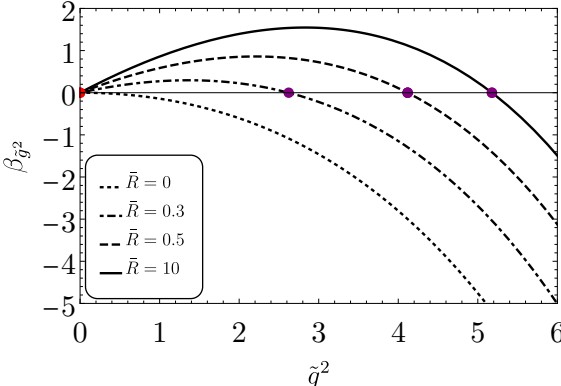

Figure 6: The one-loop beta function of $\tilde{g}^2$ as a function of $\tilde{g}^2$. The purple points stand for the nontrivial fixed point, while the red point indicates the Gaussian fixed point.

with a critical exponent evaluated by using Eq. (80) as

$$\lambda_{\tilde{u}(0)} = -\frac{\partial(\partial_t \tilde{u}(0))}{\partial \tilde{u}(0)}\bigg|_{\tilde{u}(0)=\tilde{u}_*^{\mathrm{UV}}(0)} = d_{\mathrm{eff}}(\bar{R}; 0). \tag{88}$$

Thus, the value of the critical exponent can be read off from Fig. 4. It becomes always positive for all compactification radii and thus relevant.

Finally, we show the renormalization group flow of the gauge coupling and the vacuum energy by solving Eqs. (80) and (81) numerically. To do so, we use the resumed anomalous dimension of the gauge field in Eq. (82). In the left-hand side panel of Fig. 7, the renormalization group flow of the gauge coupling is presented. We can see that for a fixed value of $\bar{R}$, the behavior of the gauge coupling changes: In the energy region $k < 1/R$, we observe the asymptotically free behavior of $\tilde{g}^2$, namely the system can be regarded as effectively a $D = 4$ gauge theory. On the other hand, in the energy region $k > 1/R$ the gauge coupling turns to an increasing behavior and then converges to the fixed point value (84). Hence, the gauge coupling is asymptotically safe and we do not suffer from any UV divergence. In high energy scales, the system behaves as a $D = 5$ gauge theory. Such a dynamical deformation of the dimension takes place thanks to the change of the effective dimension $d_{\mathrm{eff}}(\bar{R}; \theta_H)$ which arises from the threshold function (58).

Whereas the gauge coupling is finite in the deep UV regime, it diverges in the IR regime. We expect from this that in such a strong interacting regime, the gauge fields acquire a mass gap and we observe the confinement of the gauge bosons. This implies that there exits a phase transition to the confinement phase (denoted by $X$ in Table 1). The present setup however does not allow us to address confinement phenomena. For those, more elaborate methods of the functional renormalization group beyond the background approximation are required. Instead, in this work, we infer from the finite temperature Yang-Mills theory in $D = 4$ how the functional renormalization group can access the confinement phase and evaluate the gap mass of the gauge fields. We argue it in the next section.

The renormalization group flow of $\tilde{u}(0)$ is exhibited in the right-hand side panel of Fig. 7. In the deep UV regime, $\tilde{u}(0)$ converges to the nontrivial UV fixed point (87) for $\bar{R} \to \infty$, while in the IR regime, we still observe the convergence of $\tilde{u}(0)$ to the nontrivial fixed point for $\bar{R} \to 0$. The transition between the nontrivial UV fixed points takes place around the energy scale $k \sim 1/R$. We conclude now that the vacuum energy in pure Yang-Mills theories could be both UV and IR finite.

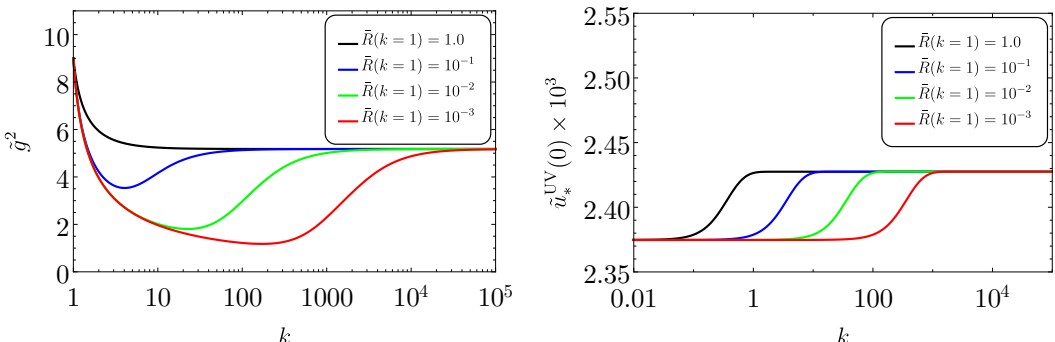

Figure 7: Flow of the gauge coupling $\tilde{g}^2$ (left) and the vacuum energy (right) for several fixed values of the dimensionless compactification radius $\bar{R}$. The energy unit of $k$ is arbitrary.

# 7 Mechanism of confinement

In the previous section, we have observed how the gauge coupling becomes strong in the IR limit. Indeed, Monte-Carlo simulations for $SU(3)$ pure Yang-Mills theory in $D = 4$ at finite temperature (corresponding to $\mathbb{R}^3 \times S^1$) show the confinement-deconfinement phase transition at a critical coupling [70]. Within the current setup, we cannot address the confinement dynamics as accounting for the dynamical emergence of a longitudinal mass for the gauge fields would require a more elaborate formulation of our setup within the FRG [71]. Here, instead, we discuss a possible mechanism of the confinement and make a small modification to the beta function of the effective potential for AB phases.

In the renormalization group evolution of $V(\theta_H) - V(0)$ which is shown in Fig. 2, its value at the origin $\theta_H = 0$ does not change from the initial value, while the potential in nonvanishing $\theta_H$ region evolutes towards larger values than the initial ones. This is because the beta function of $V(\theta_H) - V(0)$ receives positive contributions. Thus, in order for the potential of AB phases (66) to produce the nontrivial configuration $\theta_H \neq 0$ realizing the confinement phase, negative contributions to its beta function are required. In the potential (66) the gauge fields give positive contributions to the beta function, while the ghost fields contributions are negative. Hence, we may here think of two potential modifications to mimic confinement: first, enhancing the ghost-loop effects or second, suppressing the gauge-loop effects. Naively thinking, both could be realized by a large negative value of $\eta_{\mathrm{gh}} < -6$ and a large positive value of $\eta_g > 6$. However, this may be unlikely since $\eta_g$ and $\eta_{\mathrm{gh}}$ are the second order of the derivative expansion for the effective action; namely, such large values of the anomalous dimensions could imply the breakdown of the derivative expansion and the higher derivative operators cannot be ignored. This situation is thus not appropriate in the current setup or its simple extensions.

A more realistic mechanism for the suppression of the gauge fields could be considered: We could include a gap mass $m_A^2$ depending on the compactification radius in the propagator of the gauge fields which suppresses the loop effects of the gauge fields. More specifically, we modify the propagator of the gauge fields so as to be

$$\Pi(p^2) = \frac{1}{p^2 + M_{n,ij}^2 + m_A^2(R)} \,. \tag{89}$$

This modification is associated with the quantum corrections to the two-point function $\Gamma_k^{(2)}(p)$. Specifically, from the vanishing point of the full two-point function, one could read off a dressed mass $m_A$ such that $\Gamma_k^{(2)}(p = m_A) = 0$. In the strong coupling regime, it is expected that quantum

corrections to vertices become significant. In this work, however, we have used the classical propagators and vertices for the $n$-point functions ($n \geq 2$) as the background field approximation, so that such a dressed mass $m_A$ cannot be obtained. A systematic improvement for those functions can be actually realized in the vertex expansion scheme [71–96] in the functional renormalization group,[5] but this is out of the scope of the current work and then is left for future work. As a similar phenomenological approach, the Curci-Ferrari model has been studied in order to understand the center symmetry breaking in $D = 4$ YM theory at finite temperature [101–105]. In that model, the gauge-fixed action by a mass term for the gluon triggers a change of phase at small temperatures.

From the functional renormalization group analysis for a Yang-Mills theory in $D = 4$ at finite temperature [75], i.e. the gauge system in $\mathbb{R}^3 \times S^1$ where temperature $T$ corresponds to $1/(2\pi R)$ in the current setup, we emulate the behavior of the mass gap as

$$m_A^2(R) = m_D^2(R) + m_{\text{gap}}^2 \theta(k_{\text{conf}} - k), \tag{90}$$

where $\theta(x)$ is the step function: $\theta(x) \geq 1$ for $x > 0$, while $\theta(x) = 0$ for $x < 0$. Here, $k_{\text{conf}}$ is the confinement scale and $m_D$ is associated to the Debye screening mass which would naively take the form

$$m_D(R) = \frac{c_D}{2\pi R}, \tag{91}$$

as observed in finite temperature gauge theories. At the one-loop level, the Debye screening mass is obtained by the evaluation of the (non-perturbative) daisy diagram and then it turns out that the parameter $c_D$ is proportional to the gauge coupling; $c_D \sim g$. There is a prescription taking into account higher-order effects in QCD [106] and it modifies that the form of the Debye screening mass from Eq. (91). But, in this current rough sketch of the gauge-field mass, we simply treat $c_D$ as a constant parameter and set it unity: $c_D = 1$. Then, there are two free parameters: $m_{\text{gap}}^2$ represents the magnitude of the gap mass and $k_{\text{conf}}$ is the confinement scale at which the gauge fields are gapped.

Including the gap mass (90) into the propagators of the gauge fields, the flow equation for the potential $V(\theta_H)$ is given by

$$\partial_t V(\theta_H) = \frac{5k^4}{2(4\pi)^2} \widehat{\mathcal{I}}_N(\bar{R}; \theta_H; \bar{m}_A^2) - \frac{k^4}{(4\pi)^2} \mathcal{I}_N(\bar{R}; \theta_H). \tag{92}$$

Here the threshold function for the gauge fields are modified by including the gap mass so as to be

$$\widehat{\mathcal{I}}_N(\bar{R}; \theta_H; \bar{m}_A^2) = \sum_{i,j=1}^N \frac{i\pi\bar{R}}{2\sqrt{1+\bar{m}_A^2}} \left[ \cot\left( \frac{1}{2}\bar{R}\left(\theta_i - \theta_j + 2i\pi\sqrt{1+\bar{m}_A^2}\right) \right) \right.$$
$$\left. + \cot\left( -\frac{1}{2}\bar{R}\left(\theta_i - \theta_j - 2i\pi\sqrt{1+\bar{m}_A^2}\right) \right) \right], \tag{93}$$

with $\bar{m}_A^2(\bar{R}) = m_A^2(R)/k^2$ the dimensionless gap mass. Moreover, note that $\widehat{\mathcal{I}}_N(\bar{R}; \theta_H; \bar{m}_A^2 = 0) = \mathcal{I}_N(\bar{R}; \theta_H)$.

We solve the flow equation $\partial_t(V(\theta_H) - V(0))$ with the initial condition $V(\theta_H) - V(0) = 0$ and $\bar{R} = 1$ at $k = \Lambda = 10$. To represent the evolution of the potential, we set $m_{\text{gap}}^2 = 10^2$ and $k_{\text{conf}} = 9.9$. In the left-hand side panel of Fig. 8, we display the renormalization group evolution of the potential $V(\theta_H) - V(0)$ with $k_{\text{conf}} = 9.9$ at the $\theta_2 = 0$ plane. We see that

---

[5]The Schwinger-Dyson equation together with the functional renormalization group is also useful for the evaluation of the full two-point function [97–100].

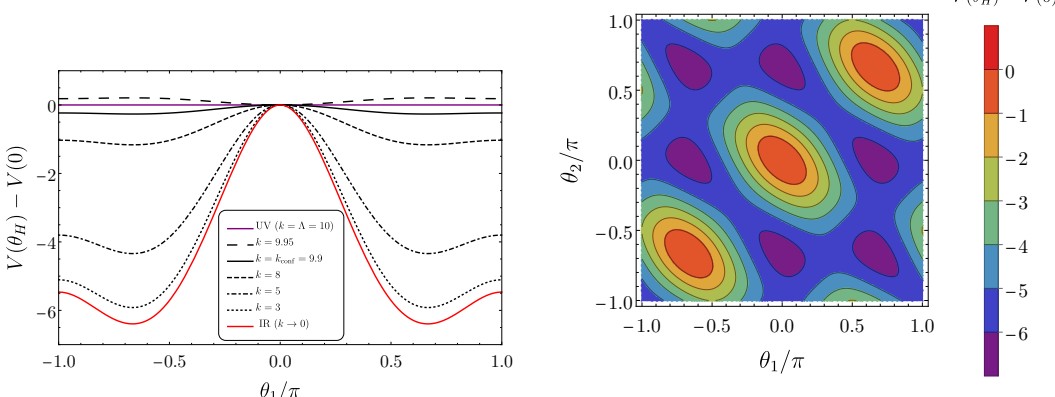

Figure 8: The potential with $m_{\text{gap}}^2 = 10^2$ and $k_{\text{conf}} = 9.9$ in a confinement-emulated scenario. Left: The renormalization evolution of the potential at the $\theta_2 = 0$ plane from $\Lambda = 10$ to $k = 0$. Right: Contour plot of the potential at the IR scale $k \to 0$ in $\theta_1$–$\theta_2$ plane.

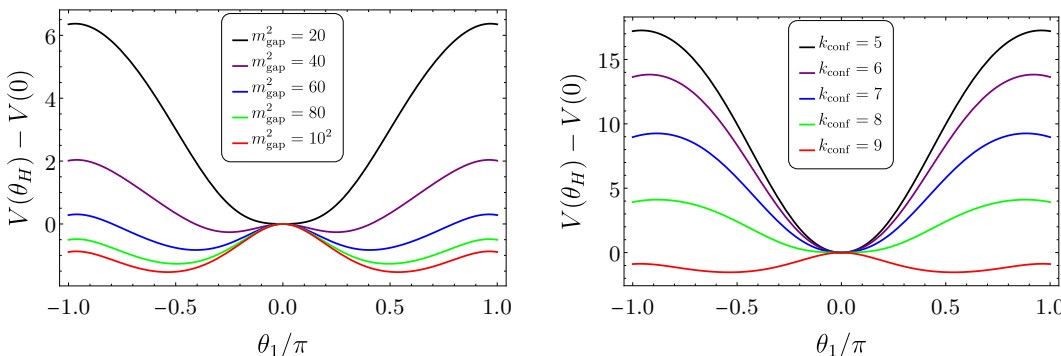

Figure 9: The IR potential at the $\theta_2 = 0$ plane in a confinement-emulated scenario. Left: Various values of $m_{\text{gap}}^2$ with $k_{\text{conf}} = 9$ fixed. Right: Various values of $k_{\text{conf}}$ with with $m_{\text{gap}}^2 = 10^2$ fixed.

the potential evolves towards positive values until $k = k_{\text{conf}} = 9.9$ at which the gap mass enters into the propagator of the gauge fields. After $k = k_{\text{conf}}$, the gauge-field contributions are suppressed and the ghost fields dominate. In the IR scale $k \to 0$, the potential converges to the red line and has nontrivial vacua at $\theta_1 = \pm 2\pi/3$ and $\theta_2 = 0$. The right-hand panel of Fig. 8 is the contour plot of $V(\theta_H) - V(0) = 0$ at $k = 0$ in $\theta_1$–$\theta_2$ plane. Minima of the potential are located at $(\theta_1, \theta_2, \theta_3) = (0, \frac{2\pi}{3}, -\frac{2\pi}{3})$ and its permutations which entails $P = 0$. Hence, this case demonstrates the potential in the confinement phase denoted by $X$. The distribution of AB phases in the confinement phase may be described by the Haar measure; see Ref. [26]. Fig. 9 exhibits the potential at the IR scale $k \to 0$ for various values of $m_{\text{gap}}^2$ and $k_{\text{conf}}$. We see that for larger $m_{\text{gap}}^2$ and $k_{\text{conf}}$, the potential tends to have confinement vacua due to enough suppressions of loop effects of the gauge fields.

# 8 Summary and Discussions

In this work, we have studied the $SU(N)$ pure Yang-Mills theory in $\mathbb{R}^4 \times S^1$ by using the functional renormalization group. We have derived the flow equations for the potential of

the AB phases and the gauge coupling. It has been shown that there exists a nontrivial UV fixed point for the gauge coupling and the vacuum energy, which entails asymptotic safety of the $SU(N)$ pure Yang-Mills theory in $\mathbb{R}^4 \times S^1$. At the nontrivial UV fixed point, those couplings are relevant parameters, i.e. free parameters in the system. We have seen also the dynamical and smooth transition of their renormalization group flows as in Fig. 7 around the energy scale of order of the inverse compactification radius. Thus, this system is UV complete and does not lose the predictivity to the low energy dynamics. These facts can be addressed only by nonperturbative treatments.

We have investigated only the two lowest order operators, i.e. $V(\theta_H)$ and $F_{MN}F^{MN}$ in the effective action; however in general an infinite number of effective operators is induced by quantum effects. In order to confirm the nonperturbative renormalizability of Yang-Mills theory in five dimensions, higher dimensional operators such as $(F_{MN}F^{MN})^2$ must be studied. Nevertheless, as discussed in Section 6, whether higher dimensional operators are still irrelevant or not relies on the magnitude of the anomalous dimension induced at the nontrivial fixed point. We expect that the magnitude of $\eta_g$ does not drastically change from the current setup since impacts of the higher dimensional operators on $\eta_g$ tend to be weaker than the lower dimensional ones. It implies that only the gauge coupling $\tilde{g}$ is a free parameter for the prediction to the low energy dynamics and all higher dimensional operators are predicted to be the nontrivial UV fixed point value in the $SU(N)$ pure Yang-Mills theory in $\mathbb{R}^4 \times S^1$. Note that the $\tilde{u}(0)$ does not play a role for the gauge dynamics in the system since it does not couple to any fields. In a gravity-matter system, however, $\tilde{u}(0)$ corresponds to the cosmological constant which couples to the metric field.[6] Thus, the relevance of the cosmological constant is significant for the prediction to the low energy dynamics in a gravity-matter system. This is part of the asymptotically safe scenario for quantum gravity [1,107,108]. See Refs. [94,109–121] for reviews. We found an IR fixed point of the cosmological constant in addition to the UV one. In a gravity-matter system, such an IR fixed point may be a key point to resolve the cosmological constant problem [122, 123]. We also note that the existence of a nontrivial fixed point in $SU(N)$ Yang-Mills theory in $D > 5$ has been discussed in Ref. [14] where a critical dimension $D_{\mathrm{cr}}$ beyond which the nontrivial fixed point disappears was found to be $5 \lesssim D_{\mathrm{cr}} < 6$.

As can be seen from the behavior of the gauge coupling in the low energy regime (see the left-hand side panel of Fig. 7), the gauge coupling becomes large. This may lead to confinement and induce a mass gap of the gauge field. Within the current setup, unfortunately we cannot address the dynamics of the confinement. To study confinement phenomena, we need improvements of the approximation. This can be systematically done within the functional renormalization group with the vertex expansion scheme [71–80, 80–96]. We can take into account vertex corrections of the gauge fields. Indeed, confinement phenomena in $D = 4$ QCD at zero and finite temperatures have been observed by using the functional renormalization group [71, 75–79]. See also Refs. [124–127]. In a future work, the same technique will be applied for $SU(N)$ pure Yang-Mills theory in $\mathbb{R}^4 \times S^1$. Instead, in this work, we have demonstrated the confinement potential by adding a gap mass $m_{\mathrm{gap}}$ of the gauge fields at a certain scale $k_{\mathrm{conf}}$ to the propagator.

Monte Carlo simulations based on lattice gauge theory are important for the elucidation of $SU(N)$ pure Yang-Mills theories. In order to remove artifacts due to a finite lattice spacing, the continuum limit is necessary. However, the existence of such continuum limit in $\mathbb{R}^4 \times S^1$ gauge theories is not obvious. Therefore, Monte-Carlo studies of the Hosotani mechanism have been

---

[6]More specifically, the cosmological constant term reads

$$\int \mathrm{d}^D x \sqrt{g}\, \tilde{u}(0)\,, \tag{94}$$

where $\sqrt{g}$ is the determinant of the metric field.

made in $\mathbb{R}^3 \times S^1$ thanks to its asymptotically free nature [26], instead of $\mathbb{R}^4 \times S^1$. The present work indicates the existence of a nontrivial UV fixed point to which the continuum limit can be taken. Indeed, the existence of such continuum limit in $SU(N)$ pure Yang-Mills theories in $\mathbb{R}^4 \times S^1$ has been explored by Refs. [128–136]. Complementary studies on higher dimensional $SU(N)$ pure Yang-Mills theories will be more important in order to elucidate the dynamics and phase structure of gauge theories.

Finally, we briefly mention the status on Gauge-Higgs unification models. A simple $SU(N)$ Yang-Mills theory in $\mathbb{R}^3 \times S^1$ is actually not realistic as a model of Gauge-Higgs unification. One of the reasons is the absence of chiral fermions in $\mathbb{R}^3 \times S^1$ spacetime. To introduce those in higher dimensional spacetime, one could consider the orbifold [137], i.e. $\mathbb{R}^3 \times S^1/Z_2$. For instance, $SU(3)$ symmetry is broken into $SU(2)_L \times U(1)_Y$ by a boundary condition of the orbifold. Then, depending on the configuration of AB phases, the $SU(2)_L \times U(1)_Y$ symmetry is further broken by the Hosotani mechanism into its subgroups such as $U(1)_{\mathrm{em}}$ and $U(1)_{\mathrm{em}} \times U(1)_Z$ [138]. Such a model however cannot reproduce the experimental results [138]. A realistic model of Gauge-Higgs unification has been built on the Randall-Sundrum warped spacetime [139] and its gauge group is given as $SO(5) \times U(1)_X$: See e.g. [140–142]. Hence, a Gauge-Higgs unification model for the standard model is constructed as a $SU(3)_c \times SO(5) \times U(1)_X$ theory; see e.g. [143, 144]. Moreover, enlarged gauge groups involving $SU(3)_c \times SO(5) \times U(1)_X$ as their subgroup have been discussed as Gauge-Higgs grand unification theories [145–151]. A UV completion for those models in the asymptotic safety scenario would be an interesting theme.

# Acknowledgements

We thank Jan M. Pawlowski and Florian Goertz for many valuable discussions and comments. M. Y. thanks also Felipe Attanasio and Kin-ya Oda for helpful discussions. This work is supported by the DFG Collaborative Research Centre "SFB 1225 (ISOQUANT)", Germany's Excellence Strategy EXC-2181/1-390900948 (the Heidelberg Excellence Cluster STRUCTURES). The work of M. Y. is supported by the Alexander von Humboldt Foundation.

# A   Heat kernel technique

In this appendix, we summarize the heat kernel expansion technique. For a review on this topic, see Refs. [152, 153]. This technique is used to derive the beta function for the gauge coupling in Appendix B.

## A.1   Heat kernel expansion

We consider the following quantity

$$\zeta = \frac{1}{2}\mathrm{tr}_{(i)}W(\Delta_i), \tag{A.1}$$

where $\Delta_i$ is the Laplacian acting on a spin-$i$ field and $\mathrm{tr}_{(i)}$ denotes the trace for the Laplacian (sum of eigenvalues of $\Delta_i$). We perform the Laplace transform

$$\mathrm{tr}_{(i)}W(\Delta_i) = \int_0^\infty \mathrm{d}s\, \widetilde{W}(s)\,\mathrm{tr}_{(i)}\left[e^{-s\Delta_i}\right]. \tag{A.2}$$

Let us here define

$$K(s,x,y) = \langle x|e^{-s\Delta_i}|y\rangle, \tag{A.3}$$

$$K(s) = \int d^d x\, K(s,x,x) = \text{tr}_{(i)}\left[e^{-s\Delta_i}\right]. \tag{A.4}$$

The kernel $K(s,x,y)$ satisfies the heat diffusion equation

$$\left(\frac{\partial}{\partial s} + \Delta_i\right)K(s,x,y) = 0, \tag{A.5}$$

with $\lim_{s\to 0} K(s,x,y) = \delta^d(x-y)$. Therefore, $K(s)$ is called "heat kernel". In particular, when the Laplacian is simply given as $-\partial^2$, its solution is

$$K(s,x,y) = \frac{\exp\left(-\frac{(x-y)^2}{4s}\right)}{(4\pi s)^{d/2}}. \tag{A.6}$$

We now consider the solution of Eq. (A.5) with a general covariant derivative. Supposing $s$ to be small, we expand $K(s,x,y)$ into a polynomial of $s$ and obtain

$$K(s,x,y) = \frac{\exp\left(-\frac{(x-y)^2}{4s}\right)}{(4\pi s)^{d/2}}\sum_{j=0} s^j \mathcal{B}_j^{(i)}(x,y), \tag{A.7}$$

from which we give

$$K(s) = K(s,x,x) = \frac{1}{(4\pi s)^{d/2}}\sum_{j=0} s^j \mathcal{A}_j^{(i)}, \tag{A.8}$$

with

$$\mathcal{A}_j^{(i)} = \int d^d x\, \mathcal{B}_j^{(i)}(x,x). \tag{A.9}$$

This expansion is the so-called "heat kernel expansion". When $\Delta = -\partial^2$, we see from Eq. (A.6) that $\mathcal{B}_0^{(i)}(x,y) = 1$ and $\mathcal{B}_j^{(i)} = 0$ for $j > 0$, and then $\mathcal{A}_j^{(i)} = \int d^d x \equiv \mathcal{V}_d$. Inserting the heat kernel expansion (A.8) into Eq. (A.2), we obtain

$$\text{tr}_{(i)} W(\Delta_i) = \frac{1}{(4\pi)^{d/2}}\sum_{j=0} Q_{\frac{d}{2}-j}[W]\,\text{tr}_{(i)}\mathcal{A}_j^{(i)} \equiv \frac{1}{(4\pi)^{d/2}}\sum_{j=0}\int d^d x\, Q_{\frac{d}{2}-j}[W] b_j^{(i)}. \tag{A.10}$$

Here, the threshold functions are defined by

$$Q_n[W] = \int_0^\infty ds\, s^{-n}\widetilde{W}(s) = \begin{cases} \dfrac{1}{\Gamma(n)}\displaystyle\int_0^\infty dz\, z^{n-1}W(z), & \text{for } n \geq 1, \\[3mm] (-1)^{-n}\dfrac{\partial^{-n}W(z)}{\partial z^{-n}}\bigg|_{z=0}, & \text{for } n \leq 0, \end{cases} \tag{A.11}$$

where the Mellin transformation is performed in the second equality. To summarize, the heat kernel expansion for the flow generator (A.1) is given by

$$\zeta = \frac{1}{2(4\pi)^{d/2}}\int d^d x \left[b_0^{(i)} Q_{\frac{d}{2}}[W] + b_1^{(i)} Q_{\frac{d}{2}-1}[W] + \cdots\right]. \tag{A.12}$$

Here, $b_n^{(i)}$ are the heat kernel coefficients. For $\Delta = -D^2 + U$, those are evaluated by

$$b_0^{(i)} = \text{tr}[I], \tag{A.13}$$

$$b_1^{(i)} = \text{tr}[U], \tag{A.14}$$

$$b_2^{(i)} = \text{tr}\left[\frac{1}{2}U^2 + \frac{1}{6}D^2U + \frac{1}{12}[D_\rho, D_\sigma][D^\rho, D^\sigma]\right], \tag{A.15}$$

where $I$ is the unity matrix. Let us now evaluate these coefficients explicitly in a Yang-Mills theory for which we take $(U)^{ab} = 2i(F_{\mu\nu})^{ab} = 2f^{abc}F_{\mu\nu}^c$ in the adjoint representation. The trace acts on Lorentz and adjoint color spaces. The heat kernel coefficients for $\Delta$ acting on spin-1 vector field are computed by

$$b_0^{(1)} = \text{tr}_{\text{Lorentz}}[I]\text{tr}_{\text{adj}}[I] = [\delta^{\mu\nu}\delta_{\mu\nu}][\delta^{ab}\delta_{ab}] = d(N^2 - 1), \tag{A.16}$$

$$b_1^{(1)} = \text{tr}_{\text{Lorentz}}\text{tr}_{\text{adj}}[2f^{abc}F^{a\mu\nu}] = 0, \tag{A.17}$$

$$\begin{aligned}
b_2^{(1)} &= \text{tr}_{\text{Lorentz}}\text{tr}_{\text{adj}}\left[\frac{1}{2}U^2 + \frac{1}{12} + \frac{1}{12}[D_\mu, D_\nu][D^\mu, D^\nu]\right] \\
&= \delta^{ab}\delta^{\mu\nu}\left[\frac{1}{2}(2f^{adc}F_{\mu\rho}^c)(2f^{dbe}F^{e\rho}{}_\nu) + \frac{\delta_{\mu\nu}}{12}(-f^{adc}F_{\rho\sigma}^c)(-f^{dbe}F^{e\rho\sigma})\right] \\
&= \frac{(24-d)N}{12}F_{\mu\nu}^a F^{a\mu\nu},
\end{aligned} \tag{A.18}$$

where we used

$$\text{tr}_{\text{adj}}(f^{abc}f^{bde}) = f^{abc}f^{bae} = N\delta^{ce}, \tag{A.19}$$

and $(T^a)_{bc} = -if^{abc}$ and $[D_\mu, D_\nu]^{ab} = -f^{abc}F_{\mu\nu}^c$ from which one finds

$$U^2 = (2iF_{\mu\rho})_{ab}(2iF^\rho{}_\nu)_{bd} = 4(F_{\mu\rho}^c T^c)_{ab}(F^{e\rho}{}_\nu T^e)_{bd} = f^{abc}f^{bde}F_{\mu\rho}^c F^{e\rho}{}_\nu, \tag{A.20}$$

$$[D_\rho, D_\sigma]^{ac}[D^\rho, D^\sigma]^{cb} = f^{abc}f^{bde}F_{\rho\sigma}^c F^{e\rho\sigma}. \tag{A.21}$$

Note $b_1^{(1)} = 0$ in Yang-Mills theories because there is no corresponding Lorentz- and gauge-invariant operator in Yang-Mills gauge theories ($\delta^{\mu\nu}F_{\mu\nu} = 0$), while in gravity the curvature scalar (Ricci scalar) is an invariant operator with the finite value of $b_1^{(1)}$. In the same manner, the heat kernel coefficients for $\Delta$ acting on spin-0 (scalar) field with $U = 0$ are found so that

$$b_0^{(0)} = N^2 - 1, \tag{A.22}$$

$$b_1^{(0)} = 0, \tag{A.23}$$

$$b_2^{(0)} = \frac{N}{12}F_{\mu\nu}^a F^{a\mu\nu}. \tag{A.24}$$

Here, we note an alternative introduction of the heat kernel [154]: One starts from writing the flow kernel as

$$\zeta = \frac{1}{2}\text{tr}_{(i)}W(\Delta_i) = \frac{1}{2}\sum_m W(\lambda_m^{(i)}), \tag{A.25}$$

where $\lambda_m^{(i)}$ are (positive) eigenvalues of the Laplacian. Using a definition of the $\delta$-function

$$\delta(z - \lambda_m^{(i)}) = \int_{a-i\infty}^{a+i\infty} \frac{ds}{2\pi i} e^{s(z - \lambda_m^{(i)})}, \tag{A.26}$$

Eq. (A.25) can be rewritten as

$$\zeta = \frac{1}{2} \sum_m \int dz \, \delta(z - \lambda_m^{(i)}) W(z) = \frac{1}{2} \sum_m \int dz \, W(z) \int_{a-i\infty}^{a+i\infty} \frac{ds}{2\pi i} e^{sz} \text{Tr}_{(i)}\left[e^{-s\Delta_i}\right], \quad (A.27)$$

where we used $\sum_m e^{-s\lambda_m^{(i)}} = \text{tr}_{(i)} e^{-s\Delta_i}$. Inserting the heat kernel expansion (A.8), we obtain again

$$\text{tr}_{(i)} W(\Delta_i) = \frac{1}{(4\pi)^{d/2}} \sum_{j=0} \int d^d x \, Q_{\frac{d}{2}-j}[W] b_j^{(i)}. \quad (A.28)$$

Here the threshold function reads in this case

$$Q_n[W] = \int dz \, W(z) \int_{a-i\infty}^{a+i\infty} \frac{ds}{2\pi i} e^{sz} s^{-n} = \frac{1}{\Gamma(n)} \int dz \, z^{n-1} W(z), \quad (A.29)$$

where we have used

$$\int_{a-i\infty}^{a+i\infty} \frac{ds}{2\pi i} e^{sz} s^{-n} = z^{n-1} \frac{i}{2\pi} \int_{\tilde{a}-i\infty}^{\tilde{a}+i\infty} d\tilde{s} \, e^{-\tilde{s}} (-\tilde{s})^{-n} = \frac{z^{n-1}}{\Gamma(n)}, \quad (A.30)$$

with $\tilde{s} = -sz$ and $\tilde{a} = -a/z$. This gives the same result as Eq. (A.11).

# B  Derivation of flow equations

We show the details of the computations for the flow equations for the potential and the gauge coupling using the heat kernel technique.

## B.1  Hessian and background field approximation

We consider fluctuations of the gauge field around a nonvanishing background $\bar{A}_M$, i.e.

$$A_M = \bar{A}_M + a_M. \quad (B.1)$$

Here and hereafter, the bar denotes the background field. Our starting effective action is

$$\Gamma_k = \frac{Z_k}{2g^2} \int d^5 x \, \text{tr} \, F_{MN} F^{MN} + \frac{Z_k}{\xi g^2} \int d^5 x \, \text{Tr}[\mathcal{F}(a)]^2 + Z_{\text{gh}} \int d^5 x \, \text{Tr}\left[\bar{c}\mathcal{M}c\right], \quad (B.2)$$

with the $R_\xi$-gauge fixing function $\mathcal{F}(a) = \delta^{\mu\nu} \bar{D}_\mu a_\nu + \xi \bar{D}_5 a_5$ and the ghost kinetic operator $\mathcal{M} = \delta^{\mu\nu} \bar{D}_\mu D_\nu + \xi \bar{D}_5 D_5$.

A central object in the functional renormalization group is the full two-point function $\Gamma_k^{(2)}$ (Hessian) which is obtained by the second functional derivative of $\Gamma_k$ with respect to fields. To get it, we evaluate the variations for the effective action

$$\Gamma_k = \Gamma_k + \delta\Gamma_k + \frac{1}{2}\delta^2\Gamma_k. \quad (B.3)$$

We need the second order variational term $\delta^2\Gamma_k$ to read off $\Gamma_k^{(2)}$. The second order variation of $A_M$ yields

$$\delta^2\Gamma_k = \frac{Z_k}{g^2} \int d^5 x \, a_M \left[(\mathcal{D}_T)^{MN} + D^M D^N\right] a_M$$

$$- \frac{Z_k}{\xi g^2} \int d^5 x \left[a_\mu \bar{D}^\mu \bar{D}^\nu a_\nu + \xi a_\mu \bar{D}^\mu \bar{D}_5 a_5 + \xi a_5 \bar{D}_5 \bar{D}^\nu a_\nu + \xi^2 a_5 (\bar{D}_5^2) a_5\right], \quad (B.4)$$

where the derivative operator is defined as

$$(\mathcal{D}_T)_{MN} = (-D_\mu^2 - D_5^2)\delta_{MN} + 2iF_{MN}\,. \tag{B.5}$$

The background field approximation enforces then

$$\Gamma_k^{(2)}[\bar{A}, A] = \Gamma_k^{(2)}[\bar{A}, A]\Big|_{\bar{A}=A} = \Gamma_k^{(2)}[\bar{A}] + S_{\text{gf}}^{(2)}[\bar{A}]\,. \tag{B.6}$$

Here, we suppose a background gauge field $\bar{A}_M$ such that $\bar{F}_{MN} = (1-\delta_{M5})(1-\delta_{N5})\bar{F}_{MN} \to \bar{F}_{\mu\nu}$ where $\bar{A}_5$ is given by Eq. (21). The mixing terms between $A_\mu$ and $a_5$ are eliminated thanks to the $R_\xi$-type gauge fixing. Hence, the Hessians for $A_\mu$ and $A_5$ are given independently as

$$(\Gamma_k^{(2)}[\bar{A}])^{\mu\nu} = \frac{Z_k}{g^2}\left\{(\bar{\mathcal{D}}_T)^{\mu\nu} - \delta^{\mu\nu}\bar{D}_5^2 + \left(1 - \frac{1}{\xi}\right)\bar{D}^\mu\bar{D}^\nu\right\}\,, \tag{B.7}$$

$$\Gamma_k^{(2)}[\bar{A}] = \frac{Z_k}{g^2}\left(\bar{\Delta} - \xi\bar{D}_5^2\right)\,, \tag{B.8}$$

with $\bar{\Delta} = -\bar{D}^2$. The Hessian for ghost fields reads from the last term in Eq. (B.2) as

$$S_{\text{gh}}^{(2)}[\bar{A}] = Z_{\text{gh}}\left(\bar{\Delta} - \xi\bar{D}_5^2\right)\,. \tag{B.9}$$

For the background $\bar{A}_5$ given in Eq. (21) and $\varphi = \varphi^a T^a = (a_\mu, a_5, c, \bar{c})$ in $\mathbb{R}^4 \times S^1$, the covariant derivative $\bar{D}_5$ turns to the KK mass spectra as

$$(\bar{D}_5\varphi)_{ij} = \partial_5\varphi_{ij} - [\bar{A}_5, \varphi]_{ij} = i\left[\frac{n}{R} - \frac{1}{2\pi R}(\theta_i - \theta_j)\right]\varphi_{ij} = iM_{n,ij}\varphi_{ij}\,. \tag{B.10}$$

The propagators, i.e. the inverse forms of Eqs. (B.7)–(B.9) are given in Eqs. (47)–(49).

We employ the regulator $\mathcal{R}_k$ such that

$$\left(\Gamma_k^{(2)} + \mathcal{R}_k(\bar{\mathcal{D}}_T)\right)_{a_\mu a_\nu}^{\mu\nu} = \frac{Z_k}{g^2}\left\{(P_k(\bar{\mathcal{D}}_T))^{\mu\nu} + \delta^{\mu\nu}M_{n,ij}^2 + \left(1 - \frac{1}{\xi}\right)\bar{D}^\mu\bar{D}^\nu\right\}\,, \tag{B.11}$$

$$\left(\Gamma_k^{(2)} + \mathcal{R}_k(\bar{\Delta})\right)_{a_5 a_5} = \frac{Z_k}{g^2}\left(P_k(\bar{\Delta}) + \xi M_{n,ij}^2\right)\,, \tag{B.12}$$

$$\left(\Gamma_k^{(2)} + \mathcal{R}_k(\bar{\Delta})\right)_{\bar{c}c} = Z_{\text{gh}}\left(P_k(\bar{\Delta}) + \xi M_{n,ij}^2\right)\,. \tag{B.13}$$

Their inverse forms correspond to the regulated propagators for $a_\mu$, $a_5$ and the ghost fields. More specifically, we find

$$\hat{\Pi}_{\mu\alpha,ij}^{(n)}(P_k) = \frac{g^2}{Z_k}\left[\frac{\delta_{\mu\nu}}{P_k + M_{ij,n}^2} + \frac{p_\mu p_\nu}{M_{ij,n}^2}\left(\frac{1}{p^2 + M_{ij,n}^2} - \frac{1}{P_k + \xi M_{ij,n}^2}\right)\right]\,, \tag{B.14}$$

$$\hat{\Pi}_{55,ij}^{(n)}(P_k) = \frac{g^2}{Z_k}\frac{1}{P_k + \xi M_{ij,n}^2}\,, \tag{B.15}$$

$$\hat{\Pi}_{\text{gh}\,ij}^{(n)}(P_k) = \frac{1}{Z_{\text{gh}}}\frac{1}{P_k + \xi M_{ij,n}^2}\,. \tag{B.16}$$

## B.2 Heat kernel expansion for Yang-Mills theory

We apply the heat kernel technique for the Yang-Mills theory in $\mathbb{R}^4 \times S^1$ spacetime. The flow equation for $\Gamma_k$ is given by

$$\partial_t\Gamma_k = \frac{1}{2}\text{Tr}_{(1)}\left[\hat{\Pi}_{\mu\alpha,ij}^{(n)}(P_k)(\partial_t\mathcal{R}_k)^{\alpha\nu}\right] + \frac{1}{2}\text{Tr}_{(0)}\left[\hat{\Pi}_{55,ij}^{(n)}(P_k)\partial_t\mathcal{R}_k\right] - \text{Tr}_{(0)}\left[\hat{\Pi}_{\text{gh}\,ij}^{(n)}(P_k)\partial_t\mathcal{R}_k\right]$$

$$= \zeta_4 + \zeta_5 + \zeta_{\text{gh}}\,. \tag{B.17}$$

We note that $\mathrm{Tr}_{(s)}$ stands for the functional trace acting on all internal spaces in which a spin-$s$ field is defined. More specifically, in the current system, the functional trace is decomposed as

$$\mathrm{Tr}_{(s)}[\mathcal{O}] = \mathrm{tr}_{\mathrm{Lorentz}} \otimes \mathrm{tr}_{\mathrm{KK}} \otimes \mathrm{tr}_{\mathrm{color}} \otimes \mathrm{tr}_{(s)}[\mathcal{O}(\bar{\Delta}_{(s)})] = \sum_{\mu=0}^{3} \delta^{\mu\nu} \sum_{n=-\infty}^{\infty} \mathrm{tr}_{\mathrm{color}} \mathrm{tr}_{(s)} \mathcal{O}_{\mu\nu;ij;n}(\bar{\Delta}_{(s)}),$$

(B.18)

where $\mathrm{tr}_{\mathrm{color}}$ acts on the internal space of $SU(N)$.

Let us first evaluate the flow generator for $A_\mu$ with the Hessian (B.7). To this end, we take the Feynman gauge $\xi = 1$. As discussed in Section 4.4, the dynamics of pure Yang-Mills theories entails the symmetric vacuum $\langle A_5 \rangle = 0$. Therefore, we can set $\theta_i = 0$, i.e. $M_{n,ij}^2 = 0$.

The flow generator for $a_\mu$ (the first term in Eq. (B.17)) reads

$$\zeta_4 = \frac{1}{2} \mathrm{Tr}_{(1)} W[\bar{\mathcal{D}}_T] = \frac{1}{2} \mathrm{Tr}_{(1)} \frac{\partial_t R_k(\bar{\mathcal{D}}_T) - \eta_g R_k(\bar{\mathcal{D}}_T)}{P_k(\bar{\mathcal{D}}_T)}$$
$$= \frac{1}{2(4\pi)^2} \int \mathrm{d}^4 x \sum_{n=-\infty}^{\infty} \left[ b_0^{(1)} Q_2[W] + b_2^{(1)} Q_0[W] \right].$$

(B.19)

Here, the heat kernel coefficients are given in Eqs. (A.16) and (A.18), and the threshold functions reads

$$Q_2[W] = \frac{1}{\Gamma(2)} \int_0^\infty \mathrm{d}z \, z \frac{\partial_t R_k(z) - \eta_g R_k(z)}{P_k(z)},$$

(B.20)

$$Q_0[W] = \frac{\partial_t R_k(z) - \eta_g R_k(z)}{P_k(z)} \bigg|_{z \to 0}.$$

(B.21)

Next, we evaluate the flow generator for $a_5$

$$\zeta_5 = \frac{1}{2} \mathrm{tr}_{(0)} W[\bar{\Delta}] = \frac{1}{2} \mathrm{tr}_{(0)} \frac{\partial_t R_k(\bar{\Delta}) - \eta_g R_k(\bar{\Delta})}{P_k(\bar{\Delta})}$$
$$= \frac{1}{2(4\pi)^2} \int \mathrm{d}^4 x \sum_{n=-\infty}^{\infty} \left[ b_0^{(0)} Q_2[W] + b_2^{(0)} Q_0[W] \right].$$

(B.22)

The heat kernel coefficients in this case are obtained in Eqs. (A.22) and (A.24) and the threshold functions are the same as in Eqs. (B.20) and (B.21).

Finally, the ghost contribution gives

$$\zeta_{\mathrm{gh}} = -\mathrm{tr}_{(0)} W_{\mathrm{gh}} = -\mathrm{tr}_{(0)} \frac{\partial_t R_k(\bar{\Delta}) - \eta_{\mathrm{gh}} R_k(\bar{\Delta})}{P_k(\bar{\Delta})}$$
$$= -\frac{1}{(4\pi)^2} \int \mathrm{d}^4 x \sum_{n=-\infty}^{\infty} \left[ b_0^{(0)} Q_2[W_{\mathrm{gh}}] + b_2^{(0)} Q_0[W_{\mathrm{gh}}] \right].$$

(B.23)

The heat kernel coefficients and the threshold functions are the same as $a_5$, while the threshold functions are given as in Eqs. (B.20) and (B.21) with the replacement $\eta_g \to \eta_{\mathrm{gh}}$.

Let us read off the beta function for the gauge coupling which is defined by the projection

$$\frac{\partial_t Z_k}{2} \frac{2\pi R}{g^2} = \frac{1}{\mathcal{V}_4} \frac{\partial^2}{\partial \bar{F}_{\mu\nu}^a \partial \bar{F}^{a\mu\nu}} \zeta.$$

(B.24)

This yields

$$
\begin{aligned}
\frac{1}{2}\frac{\partial_t Z_k}{2}\frac{2\pi R}{g^2} &= \frac{N}{2(4\pi)^2}\sum_{n=-\infty}^{\infty}\left[\frac{10}{3}Q_0[W]-\frac{1}{12}Q_0[W]+\frac{1}{6}Q_0[W_{\text{gh}}]\right] \\
&= \frac{N}{2(4\pi)^2}\left[\frac{10}{3}\left(1-\frac{\eta_g}{2}\right)-\frac{1}{6}\left(1-\frac{\eta_g}{2}\right)+\frac{1}{3}\left(1-\frac{\eta_{\text{gh}}}{2}\right)\right]\sum_{n=-\infty}^{\infty}\frac{1}{1+\left(\frac{n}{\bar{R}}\right)^2} \\
&= \frac{N}{2(4\pi)^2}\left[\frac{10}{3}\left(1-\frac{\eta_g}{2}\right)-\frac{1}{6}\left(1-\frac{\eta_g}{2}\right)+\frac{1}{3}\left(1-\frac{\eta_{\text{gh}}}{2}\right)\right]\frac{1}{N^2}\mathcal{I}_N(\bar{R};\theta_H=0),
\end{aligned}
\tag{B.25}
$$

where

$$
\frac{1}{N^2}\mathcal{I}_N(\bar{R};\theta_H=0)=\frac{1}{N^2}\sum_{i,j=1}^{N}\sum_{n=-\infty}^{\infty}\frac{1}{1+\left(\frac{n}{\bar{R}}\right)^2}=\pi\bar{R}\coth(2\pi\bar{R}).
\tag{B.26}
$$

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
