# Peer review of "UV completion of extradimensional Yang-Mills theory for Gauge-Higgs unification"

_SciPost Physics, doi:SciPost Phys. 15, 101 (2023)_

## Round 1 · Referee Report · Anonymous (Referee 1) · 2022-9-26

Report

The authors studied the SU(N) Yang-Mills (YM) theory in R^4 x S^1 by using the functional renormalization group (FRG). In particular, they derived flow equations for the potential of the Aharonov-Bohm (AB) phase and gauge coupling and showed that there is a UV fixed point for the gauge coupling constant and vacuum energy. It is clearly shown that on energy scales where the influence of extra-dimensional space is sufficiently negligible, the gauge coupling constant behaves as that in four-dimensional spacetime, while on energy scales where the influence from extra-dimensional space appears, it converges to a nontrivial value at a nontrivial UV fixed point.

The results presented in this paper are interesting and this study encourages research on gauge-Higgs unification (GHU). This study is a first step in GHU analysis by using FRG. It will be interesting to see how the non-trivial fixed points in the UV shown in this study behave in systems with additional fermions and in higher dimensions by FRG analysis. I would like to recommend this paper for publication in this journal.

---

## Round 1 · Referee Report · Anonymous (Referee 2) · 2022-9-28

Strengths

  1. Interesting and well-motivated paper
  2. Agreement with perturbation theory techniques and lattice simulations and good comparison to the literature
  3. Well written and clearly organized
  4. The appendices provide all computations

Weaknesses

  1. The phenomenological scope of the paper is a bit limited, due to the lack of fermions. This was however not the stated goal of the paper.

  2. Could the inclusion of vertex quantum corrections significantly change the conclusions of the paper, with respect to the fixed point structure ? Could the largish values for g at the fixed point invalidate the background field method?

  3. Throughout the paper $\eta_{g}$ is shown to be small and thus neglected, but this is only derived under the assumption that $\eta_{gh}$ is also small. It would be good to motivate the latter assumption.

Report

This paper studies the renormalizability of 5D Yang-Mills theory, a non-perturbative theory, by studying the fixed point structure using the functional renormalization group . The beta functions are derived and the fixed points computed under certain assumptions, indicating the existence of a non-trivial fixed point, and the RG evolution of the gauge coupling and potential is shown. A toy model for studying confinement phenomenon is used to study the potential. The paper is well written, original and very relevant for further studies on the phase structure of higher dimensional theories. I recommend the paper for publication if the requested changes/questions below are addressed.

Requested changes

  1. It would be good if points 2 and 3 under weaknesses are at least commented upon.

  2. Figure 7 seems to imply how for $\tilde{R}=0.001$ UV and IR fixed points merge, while Figure 8 right shows the flowing from IR to UV fixed point, which for $\tilde{R}=0.001$ are distinct. Isn't there a contradiction? Which also brings me to the discussion on the second column of page 13, which is a bit unclear to me, the limit $\tilde{R}\rightarrow\infty$ is not shown in that figure, isn't $k\rightarrow\infty$ meant? It's also not clear to me how eq. 86 is obtained? By taking eq. 79 equal to zero, only eq 88 is obtained.

Minor comments: 1. eq 90, shouldn't the second term read $m_{gap}^2\theta(k_{conf} -k)$? 2. the reference to eqs 49 -51 under equation 53 seems wrong. 3. in eq 37 : $(\tilde{g}_j-\tilde{g}_j)$ is a typo? 4. sentence above eq 25 "One of "its" forms"? 5. in the first paragraph of page 5, shouldn't it be $P=0$? 6. At the end of the third paragraph on page 2, fermionc->fermionic 7. eq B26 left hand side should be evaluated at $\theta_H=0$?

  • validity: good
  • significance: high
  • originality: high
  • clarity: high
  • formatting: perfect
  • grammar: perfect

Author:  Masatoshi Yamada  on 2022-12-19  [id 3150]

(in reply to Report 2 on 2022-09-28)

Weakness 1: The inclusion of fermions in the extradimensional setup is determinant for the evaluation of the phenomenological viability of these theories. Moreover, it is also needed for the phase structure study. However, this computation stands beyond the scope of the current work and is currently work in progress. It will be addressed soon in an independent study.

Weakness 2: An alternative method to the one presented in the computation of the beta function would consist in obtaining the flow of the gauge coupling from the 3-point function flow. Naively, vertex corrections would emerge at two-loop level as it is discussed below equation (79) implying a subleading correction. As the fixed-point value is not so large (g/(4π) < 1) we are not in a strongly coupled regime and therefore higher loop contributions are not dominant.

Weakness 3: As mentioned, the fixed point found is not strong enough that higher loop effects are enhanced. The coupling contribution is the dominant in ηg while the contribution of ηgh is expected to be of the same order. However the latter is suppressed by a kinematic factor 1/6 and therefore highly suppresses potential contribution. In fact, large values of ηgh or ηg may indicate a breaking of the approximation taken.

Requested change 2: We are very thankful to reviewer 2 for pointing out this unclear plot which has brought us to the correction a wrong statement in the main text. We have noticed that there are not two different solutions for the flow the potential (labelled as an IR and UV fixed points) but only two different approximations. The fixed point denoted as IR corresponds to a UV fixed point of the potential considering a vanishing gauge coupling g∗ = 0 or equivalently ηg = 0 in the D → 5 limit. Moreover, g∗ = 0 is not a UV fixed point for the gauge coupling in the D → 5 limit (see equation (83)) and therefore the approximation ηg = 0 should be discarded to avoid misunderstandings. We find this the most adequate way of proceeding although both trajectories in Figure 7 are qualitatively equivalent. To conclude, there is one single UV fixed point solution for the dimensionless potential and is the one indicated in (88). This one is obtained by considering the resumed anomalous dimension in (equation (82)) in the flow of the dimensionless potential (equation (80)).
Old Figure 7 in the last draft is now easy to understand. The line labelled as UV fixed point corresponds to a UV fixed point of the flow of the potential considering a development of a non-trivial fixed point of the gauge coupling in the D → 5 limit. Here the resumed ηg in (82) is considered in (80). This result is correct. The solution indicated as IR fixed point corresponds at a UV fixed point of the flow of the potential considering ηg = 0 and therefore g∗ = 0 in the D → 5 limit. This approximation leads to a consideration of the gaussian fixed point solution which is not present and should be discarded. Both lines meet in the D → 4 limit as there, only the gaussian fixed point is always found, independently of the approximation. Due to this, we find Figure 7 confusing and therefore opt to remove it. We remain the right panel of Figure 8 with the trajectory showing the development of the correct UV fixed point.

Minor comments: we considered and implemented all of them.

---

## Round 1 · Referee Report · Yutaka Hosotani (Referee 3) · 2022-10-1

Report

This paper analyzes pure Yang-Mills theory in $R^4 \times S^1$ by the functional renormalization group method. The result is interesting and surprising. The paper certainly is worth for publication.

Previously the model has been analyzed in perturbation theory and in the lattice simulation in $R^3 \times S^1$. Although the method/approach employed in the present paper is quite different from the previous ones, the potential $V(\theta_1, \theta_2)$ and the phase structure in the $SU(3)$ model are qualitatively similar to the previously obtained result. Furthermore the authors showed the existence of a nontrivial fixed point in the RG, which may give a deeper insight in gauge theory in higher dimensions.

It would be nice if the authors add comments on the following points.
(1) The potential $V(\theta_1, \theta_2)$ depicted in figs. 3, 9, 10, for instance, is quite similar to that in perturbation theory. Why is it? Does the functional renormalization group method contain a perturbative ingredient?
(2) The existence of a nontrivial fixed point for a finite $\bar{R}$ is highly nontrivial. How robust is it? Does the existence of matter fields (fermions) affect the result?
  • validity: -
  • significance: -
  • originality: -
  • clarity: -
  • formatting: -
  • grammar: -

Author:  Masatoshi Yamada  on 2022-12-19  [id 3149]

(in reply to Report 3 by Yutaka Hosotani on 2022-10-01)

Comment 1 : As mentioned in the reply to Report 2, the value of the non-trivial fixed point found is relatively small and therefore may allow for a perturbative treatment. The non-perturbative ingredient in 5D YM arises from the dimensionality of the coupling being negative. We have shown in the case of the gauge potential how the results are in qualitative agreement with perturbation theory as it is expected from the smallness of the coupling.

Comment 2 : Inclusion of additional matter content as fermions is expected to change the behaviour in the D=5 limit as it is very well known from D=4 YM with fermions. Therefore and to make a correct assessment of the phase structure of extradimensional extensions of the standard model, the consideration of fermions is necessary. This project is currently work in progress and hope to report on it soon. We find the FRG to be the ideal tool for this investigation as will provide us with a non-perturbative and mass dependent scheme to address the 5D limit. Also, functional methods allow for qualitatively reliable analytical computations as well as great versatility for model building and phase space investigations.

---

## Round 1 · Referee Report · Anonymous (Referee 4) · 2022-10-5

Report

The authors of the manuscript discuss the asymptotic safety scenario for YM theory in the presence of a fifth, compactified dimension. To this purpose, they use the non-perturbative renormalization group that allows them to go beyond perturbative and/or epsilon expansion considerations. Although the paper does not provide a proof that this theory is asymptotically free, the idea is interesting and the results obtained within the considered truncations show evidence that this is a relevant direction of research to be continued in the future. Overall, the paper is clearly written. However, there are certain points that need to be reconsidered before the paper can be published.

First of all, it seems pretty clear that the Hosotani mechanism is identical to the breaking of center-symmetry in YM theories at finite temperature. This equivalence should be highlighted and some appropriate references should be included (starting with the founding article by Polyakov, the lattice works by Svetitsky and collaborators, Weiss, Pisarski, Dumitru, Korthals-Altes, …).

Also, it seems that the presentation around Eqs. (8)-(12) completely misses the point, in particular about the role of center symmetry. The point is that the gauge transformations considered by the authors and which are such that V’=V in (10) are only one (trivial) subset of transformations that make A’ comply with the boundary conditions (8). More generally, one can consider transformations such that V’=VZ where Z is an element of the center of SU(N), that is Z=exp(i2pi/Nk) 1, with k=0,…,N-1. For these transformations, and if we now make the choice V=1 (as the authors did eventually) equation (11) becomes

U(x,x5+2piR) = ZU(x,x5)

These are the non-periodic transformations (more precisely, they are periodic modulo an element of the center) that change the Polyakov loop by a phase. In contrast, those transformations considered by the authors in their version of eq.(11) are periodic (once one makes the choice V=1) because this version of (11) reads

U(x,x5+2piR) = U(x,x5)

So, in no way can these transformations be non-periodic as the authors write below (12). In fact, it is pretty clear from a simple calculation that the transformation in (12) is periodic. Similarly, below (18), and contrary to what the authors write, the transformation that changes the Polyakov loop is not the one given in (12) but rather one of the transformations with Z!=1.

A related aspect which is the source of confusion in the paper is that the symmetry that is broken is not gauge symmetry, contrary to what is stated in various instances. A gauge symmetry cannot be spontaneously broken. This is quite intuitive because gauge symmetry is not physical. For a more rigorous proof one can invoke Elizur’s theorem. What happens in the present context is that the transformations satisfying U(x,x5+2piR) = ZU(x,x5) can be classified into two categories. Those with Z=1 do not change the observables and as such need to be regarded as the actual, unphysical, gauge transformations. In contrast, those with Z!=1, because they change at least one observable (the Polyakov loop) need to be regarded as physical transformations. Even though they look like gauge transformations, they are not. In particular, their breaking does not signal the breaking of gauge symmetry. More precisely, two physical transformations U1 and U2 related as U1=U2 U0 with U0 a true gauge transformation should be considered as one and the same physical transformation (in particular they transform the Polyakov loop in the same way). It follows that the actual physical symmetry group is the quotient of the group G all possible transformations with any value of Z by the group of true gauge transformations G0 with Z=1. This quotient G/G0 is isomorphic to the center of SU(N). This finite group is the actual group that is probed by the Polyakov loop and its breaking has nothing to do with the breaking of gauge symmetry.

Let me also mention that, from the point of view of the finite temperature YM theory, the phase with theta_1=theta_2=theta_3=0 or theta_1=theta_2=theta_3=pm 2pi/3 is the broken phase (in contrast to the terminology used by the authors), since there are three degenerate vacua at which the Polyakov loop acquires the values 1 and e^{pm i2pi/3}. On the other hand, the symmetric phase is the one that corresponds to a vanishing Polyakov loop. This is because if the transformations associated to Z!=1 are not broken, then the expectation value of the Polyakov loop needs to be invariant under multiplication by Z, and then it needs to vanish. The terminology used by the authors seems to be orthogonal to these considerations.

Another aspect which deserves clarification is the status of the considered truncation for the renormalization group flow. It should appear more clearly that (26) is a truncation ansatz that makes implicit assumptions. Relaxing these assumptions could have a strong impact on the results. In particular, it is well known that quadratic regulators as those used in this work break the underlying BRS symmetry of the gauge-fixed action. This implies in particular that the initial condition for Gamma_kappa at large values of kappa is not the BRS invariant classical action but should include, in principle, all new operators compatible with the remaining symmetries. For instance, all FRG studies require a gluon mass at the initial scale and thus a running mass m_k that couples a priori to Z_k and g_k. The ansatz considered by the authors completely neglects this, to date, inevitable fact. This should be stated very clearly around (26).

Regarding the gluon mass, in Sec.VII, the authors propose a phenomenological way to incorporate it in the picture, in order in particular to obtain a non-trivial Hosotani phase. I should stress that this type of phenomenological approach has been pursued with the purpose of studying center symmetry breaking (that is the same problem as considered here) of finite T d=4 YM theory (that is d=4 with a compactified dimension) using the Curci-Ferrari model which precisely amounts to supplementing the (background Landau gauge) gauge-fixed action by a mass term for the gluon. There, it is well known that the inclusion of the mass triggers a change of phase at small temperatures, due precisely to the dominance of the ghosts. The authors could consider including some of the relevant references in this respect: arXiv:1407.6469, arXiv:1511.07690, arXiv:1504.02916.

Within the same model, the one- and two-loop renormalization group flow (including the running of the gluon mass) has been considered in arXiv:1703.04041 and arXiv:1905.07262.

Minor remarks :

  • not sure that “background field approximation” is the good expression. The background field refers to a particular method, within which one can then consider various approximations;

  • in the intro, it is written “predictable”. Did the authors mean “predictive”?

  • in (31), the parameter \xi is included in the gauge-fixing functional. The (clever) reason for this choice should be mentioned here and not much later in one of the appendices.

  • in (41), one should clarify the presence of the factors 1/g^2. I guess that this is because the fields are normalized such that the action is ~F^2/g^2.

  • below (45), a more appropriate (and well known) interpretation for the covariant derivative is that it acts as a (color-dependent) chemical potential rather than a mass.

  • right before (66), the authors meant \bar A_\mu=0 and \bar A_5=0, right?

  • top of page 4, it is assumed that the vacuum state takes <A_mu>=0 and <A_5>!=0. It should be stressed however that this makes no sense prior to fixing a particular gauge. Within Landau gauge, one expects these expectation values to be both 0 due to color rotation invariance which is not broken by the gauge-fixing and is not expected to be broken spontaneously. In the present work, the possibility of finding a non-vanishing expectation value for A_5 relates to the fact that the gauge-fixing condition introduces a preferred color direction (31) via the background.

  • validity: good
  • significance: good
  • originality: good
  • clarity: good
  • formatting: good
  • grammar: good

Author:  Masatoshi Yamada  on 2022-12-19  [id 3148]

(in reply to Report 4 on 2022-10-05)

Comment 1: We agree that the gauge symmetry breaking by the Hosotani mechanism is understood as the breaking of center symmetry. We have mentioned the breaking of center-symmetry in YM theories at finite temperature and cited several references in Sec. 2.4.

Comment 2: We thank the referee very much for pointing out missing point about the residual transformation. We have modified the discussion below Eqs. (12) and (19).

Comment 3: We agree with the referee that the gauge symmetry is not spontaneously broken on the lattice thanks to Elizur’s theorem, while it is shown only when the gauge fixing is not employed. In the perturbation theory and the functional methods, a central object is the propagator for which we require the gauge fixing. The gauge symmetry breaking is observed in the sense that phase transitions take place between symmetric and broken phases. Then, the phase structure due to the Hosotani mechanism is classified by gauge symmetry breaking as listed in Table 1 in our paper. On the other hand, the lattice formulation for gauge theories does not need the gauge fixing. In this case, the phase structure is understood by the mechanism as the referee’s explanation. The two different viewpoints may be indistinguishable. Indeed, the same situation is in the standard model: In the perturbation theory and the functional methods, one observes that SU(2)×U(1) is broken into U(1) and consequently W and Z bosons become massive, while the lattice computations do not show such a symmetry breaking, but observe the finite expectation value of gauge invariant operators such as ⟨H†H⟩. The corresponding massive gauge fields are given by the composite operators such as ⟨φ†Uμφ⟩ where Uμ is the link variable. Massive spectra between Wμ and ⟨φ†Uμφ⟩ are indistinguishable. This fact indicates that there is no phase boundary between confinement and Higgs phases.
We have commented these facts in footnote 3 in page 8.

Comment 4: We agree that in terms of the Polyakov loop, configurations Ai and Ci should correspond to broken (P ! = 0) and symmetric (P = 0) phases, respectively. The terminologies in this paper rely on commutations between the Polyakov loop and SU(N) generators. We have commented this discrepancy in a footnote 4 below page 8.

Comment 5: We have commented our truncation ansatz at the end of Section 3.1.

Comment 6: In Section 7, we have mentioned the Curci-Ferrari model and have cited references.

Minor remark 1: The name, “background field approximation” in the FRG, means the truncation such that the n-point function (n > 2) is replaced to the classical vertices. This is obtained by setting Aμ = A ̄μ for the two-point function at a finite energy scale. The name “background field approximation” may be the standard jargon in the functional renormalization group (FRG) applied to gauge theories, so we would like to keep this name, but cite appropriate references.

Minor remark 2: “predictable” is correct.

Minor remark 3: We have mentioned the advantage of the gauge fixing below Eq. (31).

Minor remark 4: We have clarified the presence of the factors 1/g2 in the propagators below Eq. (41).

Minor remark 5: In hadron physics including QCD at finite temperature and density, the term M^2_{ij,m} may be called an imaginary chemical potential, while it is called a “mass” in the research area beyond the standard model.
We would like to keep calling it “mass” in this paper.

Minor remark 6: No. We consider a certain background A μ != 0 and A5 = 0, and then read off the quantum
corrections to the gauge coupling by using the heat kernel method.

Minor remark 7: The non-vanishing expectation value of A5 is obtained not by the gauge fixing, but the dynamics
of the gauge fields. In the Landau gauge, the ghost and A5 have no the mass terms as can be seen in Eqs. (48) and (49) and thus do not contribute to V (θ_H ) − V (0). However, the gauge field Aμ has the mass term even in the Landau gauge. Consequently, the non-trivial expectation value of A5 could be produced.

Anonymous on 2023-01-25  [id 3272]

(in reply to Masatoshi Yamada on 2022-12-19 [id 3148])

I thank the authors for their clarifications. Most of them answer my questions. I have still some deep disagreement regarding their answer labelled “comment 3” on whether and in which sense gauge symmetry can be broken (in the context of what is discussed in the article). Also, in my opinion, there is still some confusion between what the authors call the breaking of gauge symmetry and center-symmetry breaking. Since this is not the main point of the paper and since the other results (at vanishing background) are interesting on their own right I would be happy to accept the paper, but I would like to comment further on these questions as this may help the authors clarifying their text.

The classification of configurations given in Table I is certainly interesting and useful. However, I think that the terminology used in the text to refer to each of these cases as different breaking patterns of the gauge symmetry is misleading. It is acceptable as pure terminology of course but the problem is that it seems to imply that gauge symmetry can be broken spontaneously which it cannot in any case (of course it is broken explicitely via gauge fixing but this is not seen at the level of the observables). Even if the terminology made sense, there could not be any phase transition at some value of R in the present case between the cases A, B or C because this would mean that, on the level of the Polyakov loop (an observable) one could identify such transition and thus a breaking of gauge symmetry, in contradiction with Elitzur theorem (for instance when transitioning from A to C, the Polyakov loop would remained locked at one of the center elements).

Interestingly enough, within the Curci-Ferrari model (which amounts to adding a gluon mass to the presently discussed approach), and at least in the case of 4d YM with one compact dimension (related to temperature), a transition between C and A has been observed at one-loop order at some temperature (distinct from the center breaking temperature). However, it is believed to be an artifact as it disappears at two-loop order, see 1412.5672.

In the presence of a gluon mass, one typically finds that the system is in case C and the actual relevant question is whether it takes any arbitrary configuration within the case C or the particular center-symmetric configurations (that also belong to the case C) corresponding to the center of the red dots in Fig.3. So, if there is a distinction to be drawn, it is actually between the generic configurations in case C, and the particular center-symmetric configurations which one could call D. Center-symmetry breaking occurs (when it occurs) between D and C and has nothing to do with what the authors call gauge symmetry breaking, see for instance their paragraph starting section 2.4, where it is written “such a breaking of gauge symmetry is understood as different realizations of center symmetry”. This is incorrect I believe.

Finally, the fact that the system (in the presence of a mass) lies in the case C does not mean that gauge symmetry is broken, see the discussion in chapters 4 and 5 of 2009.04933 that can be mapped to the present one via the identification $\theta_1=(r_3+r_8/\sqrt{3})/2$ and $\theta_2=(-r_3+r_8/\sqrt{3})/2$. The potential $V(\theta_1,\theta_2)$ is invariant under G which includes in particular the invariance under G0. The group G0 is peculiar, however, for it is the group of actual gauge transformations that do not alter the physical state of the system (that is, they do not alter any of the observables, unlike G that alters at least one observable, the Polyakov loop). The interpretation of the invariance of $V(\theta_1,\theta_2)$ with respect to G0 is that the space of the variables theta is subdivided into physically equivalent regions (connected by actual gauge transformations in G0) known as Weyl chambers. For any configuration theta in one of the Weyl chambers, there are equivalent configurations within the other Weyl chambers and the whole collection of such configurations forms an orbit under G0. Now, when studying whether a given configuration is invariant or not under a particular symmetry (thus determining whether or not that configuration realizes or breaks that symmetry), what matters is how this whole orbit transforms under the symmetry. The configurations forming a G0-orbit that is invariant under some symmetry are in general not themselves invariant under that symmetry but invariant only modulo G0. In particular the center-symmetric configurations, referred to above and labelled D, are obtained by imposing the invariance under G modulo G0. Given these general considerations, one may ask whether a given configuration breaks G0 or not. Since invariance needs to be though modulo transformations in G0, the answer is clearly no, no matter which configuration one is considering. This is because, any configuration is trivially invariant under G0 modulo G0. Phrased differently, the G0-orbit of a given configuration is always invariant under G0 and thus none of the cases A, B or C (or D) should be seen as an actual breaking of G0, in line with the other observations above.

---

## Round 2 · Referee Report · Anonymous (Referee 4) · 2023-5-18

Strengths

Interesting paper regarding the asymptotic safety scenario within compact Yang-Mills theories.

Weaknesses

Misleading statements regarding "gauge symmetry breaking" and the interpretation of center-symmetry breaking, although this does not impact the main results of this work.

Report

I thank the authors for their clarifications. Most of them answer my questions. I have still some deep disagreement regarding their answer labelled “comment 3” on whether and in which sense gauge symmetry can be broken. Also, in my opinion, there is still some confusion between what the authors call the breaking of gauge symmetry and center-symmetry breaking. Since this is not the main point of the paper and since the other results (at vanishing background) are interesting on their own right I would be happy to accept the paper, but I would like to comment further on these questions as this may help the authors clarifying their text.

The classification of configurations given in Table I is certainly interesting and useful. However, I think that the terminology used in the text to refer to each of these cases as different breaking patterns of the gauge symmetry is misleading. It is acceptable as pure terminology of course but the problem is that it seems to imply that gauge symmetry can be broken spontaneously which it cannot in any case (of course it is broken explicitly via gauge fixing but this is not seen at the level of the observables).

In terms of gauge fields (which contain unphysical degrees of freedom), symmetries need always to be defined modulo gauge transformations (corresponding in the present case to transformations that are periodic at the boundaries of the compact dimension). This is crucial in particular in order to identify those configurations that comply with center symmetry because those appear as configurations that are invariant under center transformations modulo possible gauge transformations, see for instance 2009.04933. In fact this criterion applies to any physical symmetry in the problem: those configurations that comply with the physical symmetry at hand are those that are invariant under the physical transformation modulo possible gauge transformations. When applied to the particular case of gauge transformations, the criterion becomes tautological since any configuration is trivially invariant under a gauge transformation modulo a gauge transformation, see for instance 2304.00756. In this sense, any configuration complies with gauge symmetry and there is no way in which configurations of the A, B or C type could break gauge symmetry, in line with the idea (supported by Elizur theorem) that a gauge symmetry, being a mere redundancy, could not be broken in any way.

To be more specific, consider the discussion in chapters 4 and 5 of 2009.04933 that can be mapped to the present one via the identification θ1=(r3+r8/√3)/2 and θ2=(−r3+r8/√3)/2. The potential V(θ1,θ2) is invariant under G, the group of periodic transformations modulo a center element, which includes in particular the invariance under G0, the group of strictly periodic transformations. The group G0 is peculiar, however, for it is the group of actual gauge transformations that do not alter the physical state of the system (that is, they do not alter any of the observables, unlike G that alters at least one observable, the Polyakov loop). The interpretation of the invariance of V(θ1,θ2) with respect to G0 is that the space of the variables theta is subdivided into physically equivalent regions (connected by actual gauge transformations in G0) known as Weyl chambers. For any configuration theta in one of the Weyl chambers, there are equivalent configurations within the other Weyl chambers and the whole collection of such configurations forms an orbit under G0. Now, when studying whether a given configuration is invariant or not under a particular symmetry (thus determining whether or not that configuration realizes or breaks that symmetry), what matters is how this whole orbit transforms under the symmetry. The states invariant under some symmetry are those corresponding to invariant G0-orbits. Yet, the configurations that compose this invariant G0-orbit are in general not themselves invariant under that symmetry but invariant only modulo G0. In particular the center-symmetric configurations referred to above are obtained by imposing the invariance under G modulo G0. Given these general considerations, one may ask whether a given configuration breaks G0 or not. Since invariance needs to be though modulo transformations in G0, the answer is clearly no, no matter which configuration one is considering. This is because, any configuration is trivially invariant under G0 modulo G0. Phrased differently, the G0-orbit of a given configuration is always invariant under G0 and thus none of the cases A, B or C should be seen as an actual breaking of G0, in line with the other observations above.

This being said, when analyzed in the usual sense of strict invariance (that is without the “modulo a gauge transformation” decoration), the configurations of type A, B and C differ in that some of them are more invariant than the others under certain gauge transformations. If a transition occurs between any of those configurations, this can have observable consequences, although, once again, these should not be interpreted as the breaking of gauge invariance but rather as the system exploring orbits whose representatives are less invariant under certain gauge transformations (without this affecting the gauge-invariance of the orbit itself). Interestingly enough, within the Curci-Ferrari model (which amounts to adding a gluon mass to the presently discussed approach), and at least in the case of 4d YM with one compact dimension (related to temperature), a transition between C and A has been observed at one-loop order at some temperature (distinct from the center breaking temperature). However, it is believed to be an artifact as it disappears at two-loop order, see 1412.5672, at least within the range of temperatures analyzed in that reference.

In the presence of a gluon mass, one typically finds (again, within 4d compact YM) that the system is in case C and the actual relevant question is whether it takes any arbitrary configuration within the case C or the particular center-symmetric configurations (that also belong to the case C) corresponding to the center of the red dots in Fig.3. So, if there is a distinction to be drawn, it is actually between the generic configurations in case C, and the particular center-symmetric configurations which one could call D. Center-symmetry breaking occurs (when it occurs) between D and C and has nothing to do with what the authors call gauge symmetry breaking, see for instance their paragraph starting section 2.4, where it is written “such a breaking of gauge symmetry is understood as different realizations of center symmetry”. This stated connection between center-symmetry and "breaking of gauge symmetry" is incorrect I believe.
  • validity: high
  • significance: high
  • originality: good
  • clarity: ok
  • formatting: excellent
  • grammar: excellent

Author:  Masatoshi Yamada  on 2023-05-27  [id 3690]

(in reply to Report 1 on 2023-05-18)

Thank you very much for very valuable comments. We would reply to some comments. We agree that no spontaneous gauge symmetry breaking takes place thanks to the Elitzur’s theorem, while the well-known Higgs mechanism states that non-zero expectation value of the scalar field breaks the gauge symmetry. Indeed, the statement “SU(2)× U(1) gauge symmetry is broken into U(1)” is widely accepted as the standard jargon although this statement is actually incorrect in sense of the Elitzur’s theorem. The terminology in Table I is also standard in the Gauge-Higgs theories. Therefore, we would like to keep it.
We agree that a transition between C and A does not take place. Indeed, phases B and C are observed only when the adjoint fermion with finite mass is introduced. Therefore, pure YM theories may show only phases X and A where phase X originates from the distribution of the Haar measure which shows the same configuration as phase B. Thus the phase transition between X and A would be observed. See Ref. [1309.4198]
“Breaking of gauge symmetry” in this paper is classified by evaluating the commutation between the Wilson line with configuration of θi and the generators of SU(3). For instance, in phase C, only T3 and T8 within 8 generators commute with the Wilson line and then the gauge fields A3μ and A8μ are massless and the others have mass proportional to θi. We observe this fact as long as the functional method (including perturbation theory) is used although this conflicts with the Elitzur’s theorem. In order to understand this phenomenon in terms of gauge invariant quantities the Froehlich-Morchio-Strocchi mechanism should be employed, but this is out of the purpose in this paper and then we do not intend to discuss it. In order to avoid the confusion, we erase “such a breaking of gauge symmetry is understood as different realizations of center symmetry”, and write simply “As a similar system, D = 4 YM theories at finite temperature (corresponding to R3 × S1) have been discussed in Refs. [50-60]”.
We think actually that the Curci-Ferrari model in R4 ×S1 may be interesting as a Gauge- Higgs unification model and may bring a novel direction for understanding the dynamics of the Gauge-Higgs unification theory. We will consider this possibility as a future work. We would thank reviewer 4 very much again for detailed discussions.

---

## Round 2 · Author Response

Dear reviewers,
first of all, we would like to thank you for the useful and relevant comments which have triggered corrections.
Detached from all reports, we have modified the legend of Figure 8 which now indicates that the scale setting for the dimensionless compactification radius \bar R is set at k = 1.

---

## Round 2 · List of Changes

Here, we list major changes:

(i) In Section 2.2 and Section 2.3, a general discussion on the residual gauge transformation with Z!=0 was added. (See around Eqs. (12) and (13) , and below Eq. (19)).

(ii) In Section 2.4, we have commented on D=4 YM theories at finite temperature and cited several related references ([50-60]).

(iii) We have commented on the understanding of the phase structure listed in Table 1 in terms of the center symmetry in the footnote 3.

(iv) We have commented on the terminologies "broken phase" and "symmetric phase".

(v) In the end of Section 3.1, we have commented on the truncation ansatz for the effective action (27).

(vi) Old figure 7 has been removed and the right panel in new Figure 8 is plotted. Besides, the expression of the IR fixed point of the effective potential has been erased.

(vii) Below Eq. (90), we have commented on the Curci-Ferrari model with some appropriate references.

---

## Round 3 · Author Response

The draft has been modified in accordance with referee's suggestions.

---

## Round 3 · List of Changes

Erase the sentence "such a breaking of gauge symmetry is understood as different realizations of center symmetry" in page 7.

---

## Editorial Decision

published